# TransNeXt: Aggregating Diverse Attentions in One Vision Model

## Abstract

In the design of previous Vision Transformers (ViTs), different token mixers were often alternately stacked to balance the visual model's aggregation of global and local information, or to combine the characteristics of convolution with attention mechanism. In this paper, we propose **Aggregated Attention**, which is a biomimetic design-based token mixer enabling each token to have fine-grained attention to its nearest neighbor features and coarse-grained attention to global features in terms of spatial information aggregation. Furthermore, we incorporate learnable tokens that interact with conventional queries and keys, which further diversifies the generation of affinity matrices beyond merely relying on the similarity between queries and keys. All of these improvements can be achieved within a single attention layer, eliminating the need for alternately stacking different token mixers. Additionally, we propose **Convolutional GLU**, a channel mixer that bridges the gap between GLU and SE mechanism, which empowers each token to have channel attention based on its nearest neighbor image features, enhancing local modeling capability and model robustness. We combine aggregated attention and convolutional GLU to create a new visual backbone called **TransNeXt**. Extensive experiments demonstrate that our TransNeXt achieves state-of-the-art performance across multiple model sizes. At a resolution of $224^2$, TransNeXt-Tiny attains an ImageNet accuracy of **84.0%**, surpassing ConvNeXt-B with **69%** fewer parameters. Our TransNeXt-Base achieves an ImageNet accuracy of **86.2%** and an ImageNet-A accuracy of **61.6%** at a resolution of $384^2$, a COCO object detection mAP of **57.1**, and an ADE20K semantic segmentation mIoU of **54.7**.

## 1 Introduction

The Vision Transformer (ViT) (Dosovitskiy et al., 2021) has emerged as a popular backbone architecture for various computer vision tasks in recent years. The ViT model comprises two key components: the self-attention layer (token mixer) and the MLP layer (channel mixer). The self-attention mechanism plays a crucial role in feature extraction by dynamically generating an affinity matrix through similarity computations between queries and keys. This global information aggregation method has demonstrated remarkable feature extraction potential, with no inductive bias like convolution (LeCun et al., 1995), and can build powerful data-driven models. However, the transformer encoder design of vision transformers, originally developed for language modeling (Vaswani et al., 2017), exhibits inherent limitations in downstream computer vision tasks. Specifically, the computation of the global affinity matrix in self-attention poses a challenge due to its quadratic complexity and high memory consumption, which restricts its application on high-resolution image features.

In order to mitigate the computational and memory burdens imposed by the quadratic complexity inherent in the self-attention mechanism, a plethora of sparse attention mechanisms have been proposed in previous studies. One such representative method is local attention (Liu et al., 2022a), which restricts attention within a window on the feature map. However, due to the limited receptive field, this method often requires alternating stacking with different types of token mixers to achieve cross-window information exchange. Another representative method spatially downsamples the keys and values of attention (such as pooling (Wang et al., 2021a;b; Wu et al., 2021), grid sampling (Tu et al., 2022)). This method, due to its sacrifice of the query's fine-grained perception of the feature map, also has certain limitations. Recent studies (Chu et al., 2021a; Tu et al., 2022) have alternately stacked spatial downsampling attention and local attention, achieving commendable performance results.

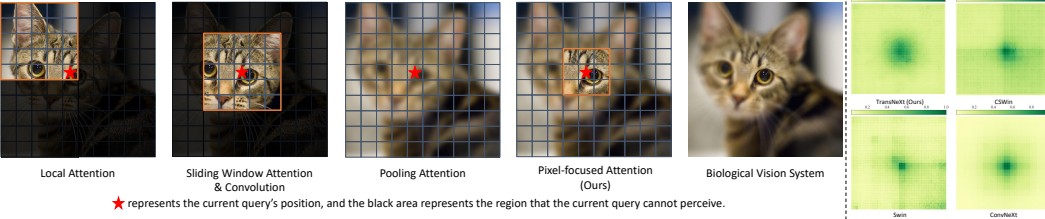

Figure 1: A comparison of prevalent visual information aggregation mechanisms, our proposed method, and biological visual systems (Left) and a visualization comparison of the Effective Receptive Field between our method and the prevalent backbone networks, using the output at stage 3 (Right).

However, recent studies (De & Smith, 2020; Veit et al., 2016) and experiments (Kim et al., 2023) have shown that deep networks with residual blocks (He et al., 2016) behave like ensembles of shallower networks, indicating that the local and global information exchange achieved by stacking blocks may not be as effective as anticipated.

On the other hand, both local attention and spatial downsampling attention differ significantly from the workings of biological vision. Biological vision possesses higher acuity for features around the visual focus and lower acuity for distant features. Moreover, as the eyeball moves, this characteristic of biological vision remains consistent for pixels at any position in the image, implying pixel-wise translational equivariance. However, for local attention based on a fixed window, tokens at the window edge and the window center are evidently not in the same situation.

We have observed that due to depth degradation effects, many efficient ViT models are unable to form sufficient information mixing through stacking. Even with a deep stack of layers, the traces of their window partitioning always form unnatural artifacts, as shown in Fig 1. To address this issue, we investigate a visual modeling approach that closely aligns with biological vision and mitigates potential model depth degradation. To this end, we initially introduce **Pixel-focused Attention**, which employs a dual-path design. In one path, each query has fine-grained attention to its nearest neighbor features, while in the other path, each query has coarse-grained attention to spatial downsampled features, allowing for a global perception. This approach operates on a per-pixel basis, effectively simulating the continuous movement of the eyeball. Furthermore, we incorporate query embedding and positional attention mechanisms into pixel-focused attention, leading to the proposal of **Aggregated Pixel-focused Attention**, which we abbreviate as **Aggregated Attention**. This approach further diversifies the generation of affinity matrices beyond merely relying on the similarity between queries and keys, thereby achieving the aggregation of multiple attention mechanisms within a single attention layer. We also reevaluate the design requirements of the channel mixer in vision transformers and propose a novel channel mixer named **Convolutional GLU**. This mixer is more apt for image tasks and integrates local feature-based channel attention to enhance model robustness.

We introduce **TransNeXt**, a hierarchical visual backbone network that incorporates **aggregated attention** as a token mixer and **convolutional GLU** as a channel mixer. Through comprehensive evaluation across image classification, object detection, and segmentation tasks, we demonstrate the efficacy of these mixing components. Our TransNeXt-Tiny, pretrained solely on ImageNet-1K, achieves an ImageNet accuracy of **84.0%**, surpassing ConvNeXt-B. In COCO object detection, it attains a box mAP of **55.1** using the DINO detection head, outperforming ConvNeXt-L pretrained at a resolution of $384^2$ by 1.7. Our TransNeXt-Small/Base, fine-tuned at a resolution of $384^2$ for merely **5 epochs**, achieves an ImageNet accuracy of **86.0%/86.2%**, surpassing the previous state-of-the-art MaxViT-Base fine-tuned for 30 epochs by 0.3%/0.5%. Moreover, when evaluated on the highly challenging ImageNet-A test set at a resolution of $384^2$, our TransNeXt-Small/Base models achieve an impressive top-1 accuracy of **58.3%/61.6%**, significantly outperforming ConvNeXt-L by 7.6%/10.9%, setting a new benchmark of robustness for ImageNet-1K supervised models.

In summary, our contributions are as follows:

1. Proposing **pixel-focused attention**, a token mixer closely aligns with biological vision and mitigates potential model depth degradation. This novel attention mechanism works on a per-pixel basis, effectively simulating the continuous movement of the eyeball and highly aligning with the focal perception mode of biological vision. It possesses visual priors comparable to convolution.

2. Proposing **aggregated attention**, an enhanced version of pixel-focused attention, which further aggregates two types of non-QKV attention mechanisms into pixel-focused attention. Notably, we propose a highly efficient approach within this framework, with the additional computational overhead accounting for a mere 0.2%-0.3% of the entire model, leading to a highly cost-effective unification of QKV attention, LKV attention, and QLV attention within a single mixer layer.

3. Proposing **length-scaled cosine attention** that enhances the extrapolation capability of existing attention mechanisms for multi-scale image input. This allows TransNeXt to achieve superior large-scale image extrapolation performance compared to pure convolutional networks B.2.

4. Proposing **convolutional GLU**, which incorporates channel attention based on nearest neighbor image features. In comparison to convolutional feed-forward, it realizes the attentionalization of the channel mixer with fewer FLOPs, thereby effectively enhancing the model's robustness.

5. Introducing **TransNeXt**, a visual backbone that delivers state-of-the-art performance in various visual tasks such as image classification, object detection, and semantic segmentation among models of similar size. It also exhibits state-of-the-art robustness.

## 2    RELATED WORK

**Vision transformers**: Vision Transformer (ViT) (Dosovitskiy et al., 2021) was the first to introduce transformer architecture to visual tasks, where images are segmented into non-overlapping patches and subsequently linearly projected into token sequences, which are later encoded by a transformer encoder. When trained with large-scale pretraining data or thoughtfully designed training strategies, ViT models outperform convolutional neural networks (CNNs)(LeCun et al., 1995; Krizhevsky et al., 2017; He et al., 2016), exhibiting remarkable performance in image classification and other downstream tasks.

**Non-QKV attention variants**: In self-attention, the dynamic affinity matrix is generated through the interaction between queries and keys. Recently, several studies (Tay et al., 2020; Li et al., 2021; Yuan et al., 2023; Arar et al., 2022) have explored the use of learnable tokens as a replacement for the original queries or keys to generate dynamic affinity matrices. Involution (Li et al., 2021) and VOLO (Yuan et al., 2023), for instance, use learnable tokens to replace the original keys, resulting in dynamic affinity matrices that are exclusively correlated with queries. In contrast, QnA (Arar et al., 2022) utilizes learnable tokens to replace queries, leading to dynamic affinity matrices that are only correlated with keys. Both methods have shown effectiveness.

**Biomimetic vision modeling**: Human vision exhibits higher acuity for features around the visual focus and lower acuity for distant features. This biomimetic design has been integrated into several machine vision models (Min et al., 2022; Yang et al., 2022; 2021). Specifically, Focal-Transformer (Yang et al., 2021) designs a visual attention based on this concept, but it operates based on window partitioning. Tokens located at the window edges cannot obtain natural foveal vision, and its window-wise manner cannot simulate the continuous movement of the human eyeball.

## 3    METHOD

### 3.1    AGGREGATED PIXEL-FOCUSED ATTENTION FOR VISION MODELS

#### 3.1.1    PIXEL-FOCUSED ATTENTION

Inspired by the functioning of biological visual systems, we have designed a pixel-focused attention mechanism that possesses fine-grained perception in the vicinity of each query, while concurrently maintaining a coarse-grained awareness of global information. To achieve the pixel-wise translational equivariance inherent in eyeball movements, we employ a dual-path design incorporating query-centered sliding window attention and pooling attention. Furthermore, to induce coupling between the two attention paths, we compute the importance in the same softmax for the query-key similarity results of both paths. This results in a competition between fine-grained and coarse-grained features, transforming pixel-focused attention into a multi-scale attention mechanism.

Given an input $X \in \mathbb{R}^{C \times H \times W}$, we now focus on the operations performed on a single pixel in the input feature map. We define a set of pixels within a sliding window centered at pixel at $(i, j)$ as $\rho(i, j)$. For a fixed window size of $k \times k$, $\|\rho(i, j)\| = k^2$. Concurrently, we define the set of pixels obtained from pooling the feature map as $\sigma(X)$. Given a pooling size of $H_p \times W_p$, $\|\sigma(X)\| = H_p W_p$. Therefore, **p**ixel-**f**ocused **a**ttention (**PFA**) can be described as follows:

$$S_{(i,j)\sim\rho(i,j)} = Q_{(i,j)}K_{\rho(i,j)}^T \quad S_{(i,j)\sim\sigma(X)} = Q_{(i,j)}K_{\sigma(X)}^T \tag{1}$$

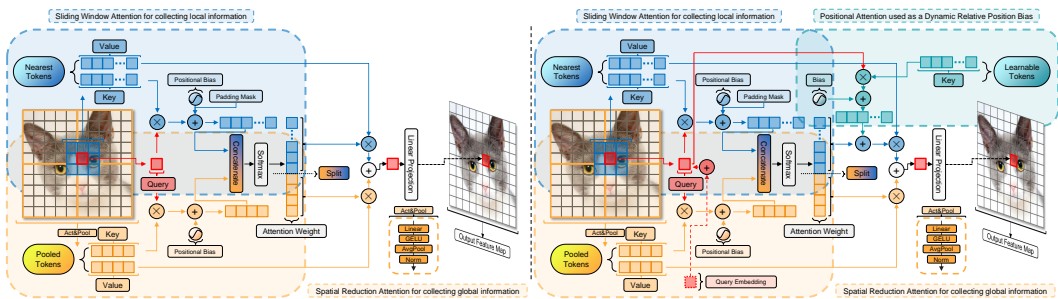

Figure 2: An illustration of the comparison between pixel-focused attention (left) and aggregated attention (right). Both have a feature size of 10×10, a window size of 3×3, and a pool size of 2×2.

$$A_{(i,j)} = \text{softmax}\left(\frac{\text{Concatenate}(S_{(i,j)\sim\rho(i,j)}, S_{(i,j)\sim\sigma(X)})}{\sqrt{d}} + B_{(i,j)}\right) \tag{2}$$

$$A_{(i,j)\sim\rho(i,j)}, A_{(i,j)\sim\sigma(X)} = \text{Split}(A_{(i,j)}) \text{ with size } [k^2, H_p W_p] \tag{3}$$

$$\mathbf{PFA}(X_{(i,j)}) = A_{(i,j)\sim\rho(i,j)}V_{\rho(i,j)} + A_{(i,j)\sim\sigma(X)}V_{\sigma(X)} \tag{4}$$

**Activate and Pool**: In order to utilize the linear complexity mode of PFA for large-scale image inference in subsequent applications, we employ parameter-free adaptive average pooling for downsampling in the spatial dimension. However, the average pooling operator significantly loses information. Therefore, we use a single-layer neural network for projection and activation before feature map pooling to compress and extract useful information in advance, thereby improving the information compression rate after downsampling. After pooling, we once again use layer normalization to normalize the output to ensure the variance consistency of $X$ and $\sigma(X)$. The downsampling operator we propose, termed 'Activate and Pool', can be expressed by the following equation:

$$\sigma(X) = \text{LayerNorm}(\text{AvgPool}(\text{GELU}(\text{Linear}(X)))) \tag{5}$$

We replaced the downsampling module in PVTv2-li (Wang et al., 2021b) with our 'activate and pool' mechanism and designed a 2M-sized model for ablation experiments on CIFAR-100 (Krizhevsky & Hinton, 2009). Our module improved the top-1 accuracy of PVTv2-li from 68.1% to 70.4%, demonstrating the effectiveness of this approach.

**Padding mask**: In the sliding window path, pixels located at the edge of the feature map inevitably compute similarities with zero-padding outside the boundary. To prevent these zero similarities from influencing the softmax operation, we employ a padding mask to set these results to $-\infty$.

### 3.1.2 AGGREGATING DIVERSE ATTENTIONS IN A SINGLE MIXER

**Query embedding**: Several vision-language models (Li et al., 2022a; 2023) utilize queries originating from the textual modality to perform cross-attention on keys derived from the visual modality, thereby achieving cross-modal information aggregation to complete Visual Question Answering (VQA) tasks. Moreover, it has been proven effective and efficient to incorporate and optimize learnable prefix query tokens when fine-tuning these multimodal models to adapt to specific subtasks.

A natural extension of this idea is to incorporate these learnable query tokens into the attention mechanism of the backbone network for well-defined tasks such as image classification, object detection, and semantic segmentation, and directly optimize them. This approach has been validated by previous work (Arar et al., 2022) for its effectiveness.

This method differs from traditional QKV attention as it does not use queries from the input but learns a query defined by the current task to perform cross-attention. Therefore, we categorize this method as **L**earnable-**K**ey-**V**alue (**LKV**) attention, drawing a parallel to QKV attention. We found that adding a learnable **Q**uery **E**mbedding (**QE**) to all query tokens in traditional QKV attention can achieve similar information aggregation effects with negligible additional overhead. We only need to modify Equation 1 as follows:

$$S_{(i,j)\sim\rho(i,j)} = (Q_{(i,j)} + \text{QE})K_{\rho(i,j)}^T \quad S_{(i,j)\sim\sigma(X)} = (Q_{(i,j)} + \text{QE})K_{\sigma(X)}^T \tag{6}$$

**Positional attention**: An alternative approach to information aggregation is the use of a set of learnable keys that interact with queries originating from the input to obtain attention weights, *i.e.*, **Q**uery-**L**earnable-**V**alue (**QLV**) attention. This method differs from traditional QKV attention as it disrupts the one-to-one correspondence between keys and values, resulting in learning more implicit relative positional information for the current query. Consequently, it is often employed in conjunction with a sliding window in visual tasks (Li et al., 2021; Yuan et al., 2023). Unlike static affinity matrices such as convolution or relative position bias, the affinity matrix generated in this way takes into account the impact of the current query and can dynamically adapt based on it. We have observed that this data-driven modeling approach exhibits greater robustness compared to static relative position bias and can further enhance locality modeling capabilities. Leveraging this feature, we introduce a set of learnable tokens $T \in \mathbb{R}^{d \times k^2}$ in each attention head, allowing these tokens to interact with queries to obtain additional dynamic position bias and add it to $A_{(i,j) \sim \rho(i,j)}$. Using this enhancement only requires an additional computational overhead of $HWk^2C$, comparable to depthwise convolution. We only need to modify Equation 4 as follows:

$$\mathbf{PFA}(X_{(i,j)}) = (A_{(i,j) \sim \rho(i,j)} + Q_{(i,j)}T)V_{\rho(i,j)} + A_{(i,j) \sim \sigma(X)}V_{\sigma(X)} \tag{7}$$

### 3.1.3 OVERCOMING MULTI-SCALE IMAGE INPUT

**Length-scaled cosine attention**: In contrast to the scaled dot product attention, the scaled cosine attention, which employs cosine similarity, has been observed to generate more moderate attention weights (Henry et al., 2020; Liu et al., 2022a) and effectively enhance the training stability of large visual models (Dehghani et al., 2023). The scaled cosine attention typically multiplies an additional learnable coefficient $\lambda$ to the cosine similarity results of queries and keys, enabling the attention mechanism to effectively ignore insignificant tokens (Henry et al., 2020). Recent studies (Hahn, 2020; Chiang & Cholak, 2022) have discovered that as the length of the input sequence increases, the confidence of the attention output decreases. Therefore, the scaling factor of the attention mechanism should be related to the length of the input sequence (Chiang & Cholak, 2022). Su (2021) further proposed that the design of attention should exhibit entropy invariance to facilitate better generalization to unknown lengths. Su (2021) provided an estimate of the entropy of the scaled dot product attention with a sequence length $n$ when queries and keys are approximated as vectors with a magnitude of $\sqrt{d}$:

$$\mathcal{H}_i \approx \log n - 0.24\lambda d + \mathcal{O}(1) \tag{8}$$

For cosine similarity, we define the queries and keys with $\ell_2$-normalization applied along their head dimensions as $\hat{Q}$ and $\hat{K}$ respectively, both of which have magnitudes of 1. To maintain entropy invariance and disregard constant terms, we set $\lambda \approx \frac{\log n}{0.24}$. Given that Equation 8 is merely an estimate, we set $\lambda = \tau \log n$, where $\tau$ is a learnable variable initialized to $\frac{1}{0.24}$ for each attention head. We propose **length-scaled cosine attention** as follows:

$$\text{Attention}(Q, K, V) = \text{softmax}(\tau \log N * \hat{Q}\hat{K}^T)V \tag{9}$$

Here, $N$ denotes the count of effective keys each query interacts with, excluding the count of masked tokens. Specifically, when applied in a transformer decoder (Vaswani et al., 2017), future tokens masked by a causal mask should not be counted in $N$. In the context of pixel-focused attention, $N$ is calculated as $N_{(i,j)} = \|\rho(i,j)\| + \|\sigma(X)\| - \|\mu(i,j)\|$, where $\mu(i,j)$ represents the set of padding-masked tokens at position $(i,j)$.

**Position bias**: To further enhance the extrapolation capability of pixel-focused attention for multi-scale image inputs, we employ different methods to calculate $B_{(i,j) \sim \rho(i,j)}$ and $B_{(i,j) \sim \sigma(X)}$ on two paths. On the pooling feature path, we use **log**-spaced **c**ontinuous **p**osition **b**ias (**log-CPB**) (Liu et al., 2022a), a 2-layer MLP with a ReLU (Nair & Hinton, 2010) to compute $B_{(i,j) \sim \sigma(X)}$ from the spatial relative coordinates $\Delta_{(i,j) \sim \sigma(X)}$ between $Q_{(i,j)}$ and $K_{\sigma(X)}$. On the sliding window path, we directly use a learnable $B_{(i,j) \sim \rho(i,j)}$. On one hand, this is because the size of the sliding window is fixed and does not require extrapolation of unknown relative position biases through log-CPB, thus saving computational resources. On the other hand, we observe that using log-CPB to calculate $B_{(i,j) \sim \rho(i,j)}$ results in performance degradation. We believe this is because $\Delta_{(i,j) \sim \sigma(X)}$ represents the spatial relative coordinates between fine-grained tokens and coarse-grained tokens,

while $\Delta_{(i,j)\sim\rho(i,j)}$ represents the spatial relative coordinates between fine-grained tokens, and their numerical meanings are different. We discuss these details further in Section B.3

**Aggregated attention**: By applying the aforementioned diverse attention aggregation methods and techniques for enhancing the extrapolation capability for multi-scale inputs, we propose an enhanced version of pixel-focused attention, termed aggregated pixel-focused attention, which we abbreviate as **A**ggregated **A**ttention (**AA**). It can be described as follows:

$$S_{(i,j)\sim\rho(i,j)} = (\hat{Q}_{(i,j)} + \text{QE})\hat{K}_{\rho(i,j)}^T \quad S_{(i,j)\sim\sigma(X)} = (\hat{Q}_{(i,j)} + \text{QE})\hat{K}_{\sigma(X)}^T \tag{10}$$

$$B_{(i,j)} = \text{Concatenate}(B_{(i,j)\sim\rho(i,j)}, \textbf{log-CPB}(\Delta_{(i,j)\sim\sigma(X)})) \tag{11}$$

$$A_{(i,j)} = \text{softmax}(\tau \log N * \text{Concatenate}(S_{(i,j)\sim\rho(i,j)}, S_{(i,j)\sim\sigma(X)}) + B_{(i,j)}) \tag{12}$$

$$A_{(i,j)\sim\rho(i,j)}, A_{(i,j)\sim\sigma(X)} = \text{Split}(A_{(i,j)}) \text{ with size } [k^2, H_p W_p] \tag{13}$$

$$\textbf{AA}(X_{(i,j)}) = (A_{(i,j)\sim\rho(i,j)} + \hat{Q}_{(i,j)}T)V_{\rho(i,j)} + A_{(i,j)\sim\sigma(X)}V_{\sigma(X)} \tag{14}$$

### 3.1.4 FEATURE ANALYSIS

**Computational complexity**: Given an input $X \in \mathbb{R}^{C\times H\times W}$, a pooling size of $H_p \times W_p$, and a window size of $k \times k$, we consider the impact of 'activate and pool' operation and linear projection. The computational complexities of pixel-focused attention and aggregated attention are:

$$\Omega(\textbf{PFA}) = 5HWC^2 + 2H_pWpC^2 + 2HWH_pW_pC + 2HWk^2C \tag{15}$$

$$\begin{aligned} \Omega(\textbf{AA}) &= \Omega(\textbf{PFA}) + HWk^2C \\ &= 5HWC^2 + 2H_pWpC^2 + 2HWH_pW_pC + 3HWk^2C \end{aligned} \tag{16}$$

We observe that when the pooling size $H_p \times W_p$ is set to a value independent of the input size, Both $\Omega(\textbf{PFA})$ and $\Omega(\textbf{AA})$ scales linearly with the length of the input sequence. This implies that both PFA and AA can perform inference in a **linear complexity mode**, as discussed in Section B.2.

**Optimal accuracy-efficiency trade-off**: Through empirical studies, we observed that the size of the sliding window has a negligible impact on model performance. Consequently, we employed the minimal form of a $3 \times 3$ sliding window to capture features near the visual focus, significantly reducing computational and memory consumption. We attribute this to the presence of pooling feature paths, which endow each query with a global receptive field, thereby greatly diminishing the need to expand the sliding window size to extend the receptive field. Detailed ablation study results and discussions can be found in Section B.5.

**Comparison with prior work**: Both PFA and AA achieve a biomimetic design that simultaneously provides fine-grained attention at the visual focus and coarse-grained attention at a distance, effectively circumventing the potential decline in effectiveness brought about by depth degradation. Compared to Focal-Transformer (Yang et al., 2021), our method works on a per-pixel basis, effectively simulating the continuous movement of the eyeball and eliminating unnatural visual perception that may be caused by partitioning windows in a fixed manner. Compared to pure convolutional networks that use large convolution kernels (Ding et al., 2022; Liu et al., 2023) to deal with potential depth degradation, our method shows significant advantages in large-scale image inference, as discussed in Section B.2.

## 3.2 CONVOLUTIONAL GLU

### 3.2.1 MOTIVATION

**Gated channel attention in ViT era**: Previous work, represented by the Squeeze-and-Excitation (SE) mechanism (Hu et al., 2020), first introduced channel attention into the field of computer vision, which uses a branch with an activation function to gate the network output. In gated channel attention, the gating branch has more decision-making power than the value branch, and it ultimately determines whether the corresponding output elements are zeroed. From this perspective, the SE mechanism cleverly uses features after global average pooling as the input of the gating branch, achieving a largest receptive field for better decision-making and solving the problem of insufficient receptive

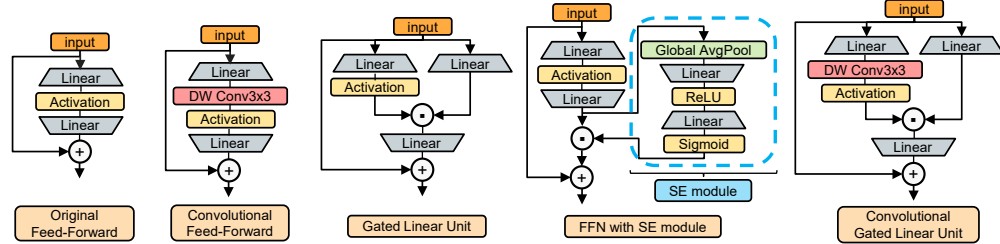

Figure 3: Comparison of prevalent channel mixer designs and Convolutional GLU

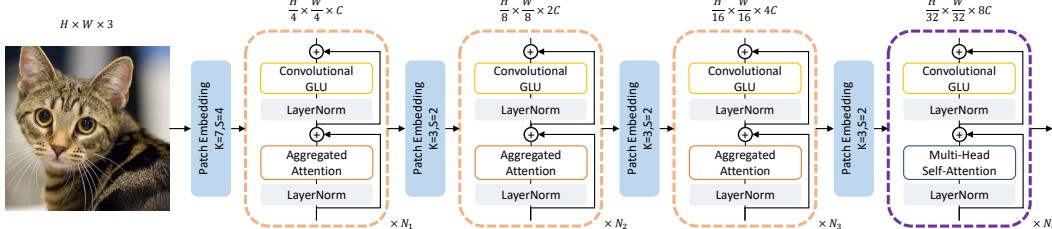

Figure 4: An illustration of TrasnNeXt architecture.

field in CNNs structures at the same time. However, in the ViT era, global receptive fields are no longer scarce. Various global token mixers represented by self-attention have achieved higher quality global information aggregation than global average pooling. This makes the global pooling method used by the SE mechanism show some shortcomings, such as this method makes all tokens on the feature map share the same gating signal, making its channel attention lack flexibility and too coarse-grained. Despite this, it's worth noting that ViT structures lack channel attention. Recent research (Zhou et al., 2022) has found that incorporating the SE mechanism into a channel mixer can effectively enhance model robustness, as shown in Fig. 3.

**Convolution in ViT era**: Recent studies (Chu et al., 2021b; Islam et al., 2020) have shown that introducing a $3 \times 3$ depthwise convolution (Chollet, 2017) into the vision transformer can be viewed as a form of conditional position encoding (CPE) (Chu et al., 2021b), which effectively captures positional information from zero-padding.

### 3.2.2 RETHINKING CHANNEL MIXER DESIGN

The Gated Linear Unit (GLU) (Dauphin et al., 2017; Shazeer, 2020) is a channel mixer that has been shown to outperform Multi-Layer Perceptron (MLP) in various natural language processing tasks. GLU consists of two linear projections that are element-wise multiplied, with one projection being activated by a gating function. Unlike the SE mechanism, its gating signal for each token is derived from the token itself and does not have a larger receptive field than the value branch.

**More elegant design**: We found that simply adding a minimal form of $3 \times 3$ depthwise convolution before the activation function of GLU's gating branch can make its structure conform to the design concept of gated channel attention and convert it into a gated channel attention mechanism based on nearest neighbor features. We named this method **Convolutional GLU**, as shown in Fig. 3.

**Feature analysis**: **Conv**olutional **GLU** (**ConvGLU**) addresses the overly coarse-grained drawback of the SE mechanism, where all value tokens share the same gating signal. It also meets the needs of some ViT models without position encoding design that require position information provided by depthwise convolution. Moreover, the value branch of this design still maintains the same depth as MLP and GLU, making it backpropagation-friendly. When keeping the parameter volume consistent with the Convolutional Feed-Forward (ConvFFN) (Wang et al., 2021b) with an expansion ratio of $R$ and a convolution kernel size of $k \times k$, the computational complexity of ConvGLU is $2RHWC^2 + \frac{2}{3}RHWCk^2$, which is less than the $2RHWC^2 + RHWCk^2$ of ConvFFN. These characteristics make ConvGLU a promising token mixer for vision model in ViT era.

### 3.3 ARCHITECTURE DESIGN OF TRANSNEXT

In order to ensure consistency in subsequent ablation experiments B.1, TransNeXt adopts the same four-stage hierarchical backbone and overlap patch embedding as PVTv2 (Wang et al., 2021b). The

pooling feature size of the aggregated attention in stages 1-3 is also set to $\frac{H}{32} \times \frac{W}{32}$, identical to PVTv2. In stage 4, as the feature map size has been reduced to $\frac{H}{32} \times \frac{W}{32}$, the feature pooling module cannot function properly. We employ a modified version of multi-head self-ttention (MHSA) that applies query embedding and length-scaled cosine attention. This is consistent with PVTv2's use of MHSA in the fourth stage. For the channel mixer in stages 1-4, we use convolutional GLU with GELU (Hendrycks & Gimpel, 2016) activation. The expansion ratio also follows PVTv2's [8,8,4,4] setting. To ensure consistency with typical MLP parameters, the hidden dimension of convolutional GLU is $\frac{2}{3} \times$ of the set value. Furthermore, we set the head dimension to be 24 for divisibility by 3 in the channel dimension. The specific configurations of TransNeXt variants can be found in Table 2.

## 4 EXPERIMENT

| Model | #Params. (M) | FLOPs (G) | IN-1K ↑ Top-1(%) | IN-C ↓ mCE(%) | IN-A ↑ Top-1(%) | IN-R ↑ Top-1(%) | Sketch ↑ Top-1(%) | IN-V2 ↑ Top-1(%) |
|---|---|---|---|---|---|---|---|---|
| **ImageNet-1K $224^2$ pre-trained models** | | | | | | | | |
| PVT-Tiny (Wang et al., 2021a) | 13.2 | 1.9 | 75.1 | 79.6 | 8.2 | 33.7 | 21.3 | 63.0 |
| PVTv2-B1 (Wang et al., 2021b) | 14.0 | 2.1 | 78.7 | 62.6 | 14.7 | 41.8 | 28.9 | 66.9 |
| BiFormer-T (Zhu et al., 2023) | 13.1 | 2.2 | 81.4 | 55.7 | 25.7 | 45.4 | 31.5 | 70.6 |
| EfficientFormerv2-S2 (Li et al., 2022c) | 12.7 | 1.3 | 81.6 | – | – | – | – | – |
| **TransNeXt-Micro (Ours)** | 12.8 | 2.7 | **82.5** | **50.8** | **29.9** | **45.8** | **33.0** | **72.6** |
| DeiT-Small/16 (Touvron et al., 2021) | 22.1 | 4.6 | 79.9 | 54.6 | 19.8 | 41.9 | 29.1 | 68.4 |
| Swin-T (Liu et al., 2021) | 28.3 | 4.5 | 81.2 | 62.0 | 21.7 | 41.3 | 29.0 | 69.7 |
| PVTv2-B2 (Wang et al., 2021b) | 25.4 | 4.0 | 82.0 | 52.6 | 27.9 | 45.1 | 32.8 | 71.6 |
| ConvNeXt-T (Liu et al., 2022b) | 28.6 | 4.5 | 82.1 | 53.2 | 24.2 | 47.2 | 33.8 | 71.0 |
| Focal-T (Yang et al., 2021) | 29.1 | 4.9 | 82.2 | – | – | – | – | – |
| FocalNet-T (LRF) (Yang et al., 2022) | 28.6 | 4.5 | 82.3 | 55.0 | 23.5 | 45.1 | 31.8 | 71.2 |
| MaxViT-Tiny (Tu et al., 2022) | 30.9 | 5.6 | 83.4 | 49.6 | 32.8 | 48.3 | 36.3 | 72.9 |
| BiFormer-S (Zhu et al., 2023) | 25.5 | 4.5 | 83.8 | 48.5 | 39.5 | 49.6 | 36.4 | 73.7 |
| **TransNeXt-Tiny (Ours)** | 28.2 | 5.7 | **84.0** | **46.5** | **39.9** | **49.6** | **37.6** | **73.8** |
| Swin-S (Liu et al., 2021) | 49.6 | 8.7 | 83.1 | 54.9 | 32.9 | 44.9 | 32.0 | 72.1 |
| ConvNeXt-S (Liu et al., 2022b) | 50.2 | 8.7 | 83.1 | 49.5 | 31.3 | 49.6 | 37.1 | 72.5 |
| PVTv2-B3 (Wang et al., 2021b) | 45.2 | 6.9 | 83.2 | 48.0 | 33.3 | 49.2 | 36.7 | 73.0 |
| Focal-S (Yang et al., 2021) | 51.1 | 9.1 | 83.5 | – | – | – | – | – |
| FocalNet-S (LRF) (Yang et al., 2022) | 50.3 | 8.7 | 83.5 | 51.0 | 33.8 | 47.7 | 35.1 | 72.7 |
| PVTv2-B4 (Wang et al., 2021b) | 62.6 | 10.1 | 83.6 | 46.5 | 37.1 | 49.8 | 37.5 | 73.5 |
| BiFormer-B (Zhu et al., 2023) | 56.8 | 9.8 | 84.3 | 47.2 | 44.3 | 49.7 | 35.3 | 74.0 |
| MaxViT-Small (Tu et al., 2022) | 68.9 | 11.7 | 84.4 | 46.4 | 40.0 | 50.6 | 38.3 | 74.0 |
| **TransNeXt-Small (Ours)** | 49.7 | 10.3 | **84.7** | **43.9** | **47.1** | **52.5** | **39.7** | **74.8** |
| DeiT-Base/16 (Touvron et al., 2021) | 86.6 | 17.6 | 81.8 | 48.5 | 28.1 | 44.7 | 32.0 | 70.9 |
| Swin-B (Liu et al., 2021) | 87.8 | 15.4 | 83.5 | 54.5 | 35.9 | 46.6 | 32.4 | 72.3 |
| PVTv2-B5 (Wang et al., 2021b) | 82.0 | 11.8 | 83.8 | 45.9 | 36.8 | 49.8 | 37.2 | 73.4 |
| Focal-B (Yang et al., 2021) | 89.8 | 16.0 | 83.8 | – | – | – | – | – |
| ConvNeXt-B (Liu et al., 2022b) | 88.6 | 15.4 | 83.8 | 46.8 | 36.7 | 51.3 | 38.2 | 73.7 |
| FocalNet-B (LRF) (Yang et al., 2022) | 88.7 | 15.4 | 83.9 | 49.5 | 38.3 | 48.1 | 35.7 | 73.5 |
| **TransNeXt-Base (Ours)** | 89.7 | 18.4 | **84.8** | **43.5** | **50.6** | **53.9** | **41.4** | **75.1** |
| MaxViT-Base (Tu et al., 2022) | 119.5 | 24.0 | **84.9** | 43.6 | 44.2 | 52.5 | 40.1 | 74.5 |
| **ImageNet-1K $384^2$ fine-tuned models** | | | | | | | | |
| Swin-B (Liu et al., 2021) | 87.8 | 47.1 | 84.5 | – | 42.0 | 47.2 | 33.4 | 73.2 |
| ConvNeXt-B (Liu et al., 2022b) | 88.6 | 45.2 | 85.1 | – | 45.6 | 52.9 | 39.5 | 75.2 |
| MaxViT-Small (Tu et al., 2022) | 68.9 | 36.1 | 85.2 | – | 48.3 | – | – | – |
| ConvNeXt-L (Liu et al., 2022b) | 197.8 | 101.1 | 85.5 | – | 50.7 | 54.6 | 41.0 | 76.0 |
| MaxViT-Base (Tu et al., 2022) | 119.5 | 74.2 | 85.7 | – | 55.1 | – | – | – |
| **TransNeXt-Small (Ours)** | 49.7 | 32.1 | **86.0** | – | **58.3** | **56.4** | **43.2** | **76.8** |
| **TransNeXt-Base (Ours)** | 89.7 | 56.3 | **86.2** | – | **61.6** | **57.7** | **44.7** | **77.0** |

Table 1: A comprehensive comparison on the ImageNet-1K classification and additional robustness test sets.

**ImageNet-1K classification**: Our code is implemented based on PVTv2 (Wang et al., 2021b) and follows the DeiT (Touvron et al., 2021) recipe for training. The model is trained from scratch on the ImageNet-1K (Deng et al., 2009) dataset for 300 epochs, leveraging automatic mixed precision (AMP) across $8\times$ GPUs. The specific hyperparameters employed during training are detailed in Table 3. To conduct a comprehensive evaluation of the model's robustness, we utilize several additional test sets. These include ImageNet-C (Hendrycks & Dietterich, 2019), a $224^2$-sized test set that applies algorithmic distortions to ImageNet-1K validation set; ImageNet-A (Hendrycks et al., 2021), a test set comprising adversarial examples; ImageNet-R (Hendrycks & Dietterich, 2019), an extended test set containing samples that ResNet-50 (He et al., 2016) failed to classify correctly; ImageNet-Sketch (Wang et al., 2019), which contains hand-drawn images; and ImageNet-V2 (Recht et al., 2019), an extended test set that employs the same sampling strategy as ImageNet-1K.

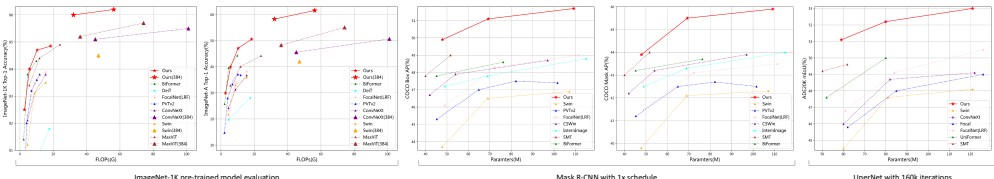

Figure 5: A comprehensive comparison of performance on ImageNet-1K, robustness on ImageNet-A, COCO detection and instance segmentation performance based on Mask R-CNN $1\times$, ADE20K semantic segmentation performance based on UperNet.

**Experimental results**: The experimental results, presented in Table 1, establish that our proposed model sets a new benchmark in ImageNet-1K accuracy and robustness across various scales. Specifically, our TransNeXt-Micro model achieves a top-1 accuracy of **82.5%** on ImageNet-1K, surpassing the FocalNet-T(LRF) while utilizing 55% fewer parameters. Similarly, our TransNeXt-Tiny model achieves a top-1 accuracy of **84.0%**, outperforming ConvNeXt-B with a reduction of 69% in parameters. Remarkably, at a resolution of $384^2$, our TransNeXt-Small/Base model surpasses the larger MaxViT-Base model by **0.3%/0.5%** respectively after only **5 epochs** of fine-tuning, compared to the 30 epochs used by MaxViT-Base. In terms of robustness, our model exhibits superior performance on five additional test sets. Notably, on the most challenging ImageNet-A test set, TransNeXt demonstrates a significant advantage in robustness as the model scales up. On ImageNet-A at a resolution of $224^2$, our TransNeXt-Base surpasses MaxViT-Base by 6.4%. At a resolution of $384^2$, our TransNeXt-Small/Base achieves an impressive ImageNet-A accuracy of **58.3%/61.6%**, significantly outperforming ConvNeXt-L by 7.6%/10.9%, while their parameter counts are only 25% and 45% of ConvNeXt-L, respectively.

**Object detection and instance segmentation**: We employed a Mask R-CNN (He et al., 2020) detection head, trained under a $1\times$ schedule, to evaluate the performance of ImageNet-1K pretrained TransNeXt on object detection and instance segmentation on the COCO (Lin et al., 2014) dataset. The experimental results are presented in Fig 5 and Table 12. Our model demonstrated comprehensive superiority when compared with previous state-of-the-art models. Notably, even our tiny model surpassed the base models of FocalNet, InternImage and CSWin in terms of $AP^b$. Similarly, we utilized a DINO (Zhang et al., 2023) detection head, also trained under a $1\times$ schedule, to further assess the potential of our model for object detection. The results can be found in Table 13. Our TransNeXt-Tiny model achieved an $AP^b$ of 55.1 under a 4-scales setting, surpassing ConvNeXt-L 1.7 with only 14% of the latter's backbone parameters. Our TransNeXt-Base achieved an $AP^b$ of 57.1 under a 5-scales setting, approaching the performance of Swin-L pretrained on ImageNet-22K.

**Semantic segmentation**: We used UperNet (Xiao et al., 2018) and Mask2Former (Cheng et al., 2022) methods to train the ImageNet-1K pretrained TransNeXt at a resolution of $512^2$ for 160k iterations, and evaluated its semantic segmentation performance on ADE20K (Zhou et al., 2019). The results from UperNet are shown in Fig 5 and Table 14, while those from Mask2Former are in Table 15. Under the UperNet method, as shown in Fig 5, our TransNeXt demonstrated comprehensive superiority over previous methods across all sizes. Our TransNeXt-Base even surpassed ConvNeXt-B, which was pretrained on ImageNet-22K and further trained at a resolution of $640^2$. Similarly, under the Mask2Former method, our TransNeXt-Small achieved an mIoU of 54.1, surpassing Swin-B which was pretrained on ImageNet-22K and further trained at a resolution of $640^2$. Furthermore, our TransNeXt-Base achieved an mIoU of 54.7. These results indicate that our method has the potential to transcend model size limitations and break through data volume barriers.

## 5 CONCLUSION

In this work, we propose a biomimetic design-based token mixer, **Aggregated Attention**, and a channel mixer with gated channel attention, **Convolutional GLU**. We combine them to propose a powerful and highly robust visual model, **TransNeXt**, which achieves state-of-the-art performance in various visual tasks such as classification, detection, and segmentation. Extensive experiments validate the effectiveness and generality of our approach. Furthermore, we provide a CUDA implementation B.7 that achieves up to 103.4% acceleration in training and 60.5% acceleration in inference.

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

# A  DETAILED SETTINGS

## A.1  CONFIGURATIONS OF TRANSNEXT VARIANTS

| Model | Channels | Head dims | Blocks | MLP ratio | Token mixer | Window size | Pool size |
|---|---|---|---|---|---|---|---|
| TransNeXt-Micro | [48, 96, 192, 384] | 24 | [2, 2, 15, 2] | [8, 8, 4, 4] | **A-A-A-M** | [3, 3, 3, −] | [7, 7, 7, −] |
| TransNeXt-Tiny | [72, 144, 288, 576] | 24 | [2, 2, 15, 2] | [8, 8, 4, 4] | **A-A-A-M** | [3, 3, 3, −] | [7, 7, 7, −] |
| TransNeXt-Small | [72, 144, 288, 576] | 24 | [5, 5, 22, 5] | [8, 8, 4, 4] | **A-A-A-M** | [3, 3, 3, −] | [7, 7, 7, −] |
| TransNeXt-Base | [96, 192, 384, 768] | 24 | [5, 5, 23, 5] | [8, 8, 4, 4] | **A-A-A-M** | [3, 3, 3, −] | [7, 7, 7, −] |

Table 2: The configurations of TransNeXt variants. The value of pool size is calculated at $224^2$ resolution. **A** = aggregated attention, while **M** = multi-head self-attention.

## A.2  TRAINING SETTINGS FOR IMAGENET-1K

To ensure reproducibility and consistency with prior work, we adopt the training strategy of PVTv2 (Wang et al., 2021b), which incorporates various data augmentation techniques, including Random Augmentation (Cubuk et al., 2020), Mixup (Zhang et al., 2018), CutMix (Yun et al., 2019), and Random Erasing (Zhong et al., 2020). To regularize our model, we employ Label Smoothing (Szegedy et al., 2016) and DropPath (Huang et al., 2016). We optimize our model using AdamW (Loshchilov & Hutter, 2019) optimizer with a gradient clipping norm of 1.0 and a weight decay of 0.05. The initial learning rate for all models is set to $10^{-3}$, with a warm-up period of 5 epochs and an initial warm-up learning rate of $10^{-6}$. We utilize the cosine learning rate scheduler (Loshchilov & Hutter, 2017) to decay the learning rate. During training, we randomly crop images to a size of $224 \times 224$. During the evaluation phase, for images with a resolution less than $384 \times 384$, we apply a center-crop with a crop ratio of 0.875. However, for images of larger sizes, we do not perform any cropping, following previous work (Liu et al., 2022b). We do not employ the EMA weights. The stochastic depth drop rates for each model are provided in Table 3.

| dataset | ImageNet-1K | |
|---|---|---|
| configuration
task | TransNeXt-Micro/Tiny/Small/Base
$224^2$ Pre-training | TransNeXt-Small/Base
$384^2$ Fine-tuning |
| batch size | 1024 | 1024 |
| base learning rate | 1e-3 | 1e-5 |
| learning rate scheduler | cosine | constant |
| min learning rate | 1e-5 | 1e-5 |
| training epochs | 300 | 5 |
| warm-up epochs | 5 | None |
| warm-up schedule | linear | None |
| warm-up learning rate | 1e-6 | None |
| optimizer | AdamW | AdamW |
| optimizer momentum | $\beta_1, \beta_2 = 0.9, 0.999$ | $\beta_1, \beta_2 = 0.9, 0.999$ |
| color jitter factor | 0.4 | 0.4 |
| auto-aug | rand-m9-mstd0.5-inc1 | rand-m9-mstd0.5-inc1 |
| random-erasing prob. | 0.25 | 0.25 |
| random-erasing mode | pixel | pixel |
| mixup $\alpha$ | 0.8 | 0.8 |
| cutmix $\alpha$ | 1.0 | None |
| mixup prob. | 1.0 | 1.0 |
| mixup switch prob. | 0.5 | 0.5 |
| stochastic drop path rate | 0.15/0.25/0.45/0.6 | 0.7/0.8 |
| label smoothing | 0.1 | 0.1 |
| gradient clip | 1.0 | 1.0 |
| weight decay | 0.05 | 0.05 |
| exp. mov. avg. (EMA) | None | None |

Table 3: The pre-training and fine-tuning settings of TransNeXt on ImageNet-1K (Deng et al., 2009).

### A.3 TRAINING SETTINGS FOR DOWNSTREAM TASKS

For experiments on the ADE20K (Zhou et al., 2019) and COCO (Lin et al., 2014) datasets, we followed the training settings of Swin (Liu et al., 2021). We utilized the MMDetection (Chen et al., 2019) and MMSegmentation (Contributors, 2020) toolboxes for training.

For the COCO 2017 dataset (Lin et al., 2014), we configured the learning rate to $10^{-4}$ and the weight decay to 0.05. In the context of the Mask R-CNN and DINO methods, the stochastic depth drop rates for TransNeXt-Tiny, TransNeXt-Small, and TransNeXt-Base were set to 0.3, 0.5, and 0.6, respectively. The model was trained for 12 epochs with a batch size of 16 using the standard $1\times$ schedule.

For the ADE20K dataset (Zhou et al., 2019), in the UperNet method, we set the learning rate to $6 \times 10^{-5}$ and the weight decay to 0.05. The stochastic depth drop rates for TransNeXt-Tiny, TransNeXt-Small, and TransNeXt-Base were set to 0.4, 0.6, and 0.7, respectively. For the Mask2Former method, we set the learning rate to $10^{-4}$ and the weight decay to 0.05, with the stochastic depth drop rates for TransNeXt-Tiny, TransNeXt-Small, and TransNeXt-Base set to 0.3, 0.5, and 0.6 respectively. All models were trained for 160K iterations with a batch size of 16 on the ADE20K dataset.

## B ABLATION STUDY

### B.1 A ROADMAP FROM PVT TO TRANSNEXT

| Step | Method | #Params. (M) | FLOPs (G) | IN-1K ↑ Top-1(%) | IN-C ↓ mCE(%) | IN-A ↑ Top-1(%) | IN-R ↑ Top-1(%) | Sketch ↑ Top-1(%) | IN-V2 ↑ Top-1(%) |
|---|---|---|---|---|---|---|---|---|---|
| 0 | PVT-Tiny (Wang et al., 2021a) | 13.2 | 1.9 | 75.1 | 79.6 | 8.2 | 33.7 | 21.3 | 63.0 |
| 1 | PVTv2-B1 (Wang et al., 2021b) | 14.0 | 2.1 | 78.7 (+3.6) | 62.6 (+17.0) | 14.7 (+6.5) | 41.8 (+8.1) | 28.9 (+7.6) | 66.9 (+3.9) |
| 2 | Deeper and Thinner | 14.9 | 2.3 | 80.08 (+1.38) | 55.3 (+7.3) | 19.7 (+5.0) | 43.2 (+1.4) | 31.1 (+2.2) | 69.2 (+2.3) |
| 3 | + More Heads | 14.9 | 2.3 | 80.12 (+0.04) | 55.0 (+0.3) | 19.2 (-0.5) | 43.5 (+0.3) | 31.5 (+0.4) | 69.4 (+0.2) |
| 4 | ConvFFN→GLU | 14.8 | 2.2 | 79.7 (-0.42) | 59.5 (-4.5) | 18.9 (-0.3) | 39.3 (-4.2) | 26.8 (-4.7) | 69.0 (-0.4) |
| 5 | GLU→ConvGLU | 14.9 | 2.2 | 80.9 (+1.2) | 54.6 (+4.9) | 23.5 (+4.6) | 44.3 (+5.0) | 32.7 (+5.9) | 70.6 (+1.6) |
| 6 | SRA→PFA | 12.8 | 2.7 | 81.8 (+0.9) | 51.7 (+2.9) | 26.9 (+3.4) | 45.2 (+0.9) | 33.3 (+0.6) | 71.6 (+1.0) |
| 7 | + Positional Attention | 12.8 | 2.7 | 82.2 (+0.4) | **50.7** (+1.0) | **31.0** (+4.1) | **46.4** (+1.2) | **34.1** (+0.8) | 72.0 (+0.4) |
| 8 | + Query Embedding | 12.8 | 2.7 | **82.5** (+0.3) | 50.8 (-0.1) | 29.9 (-1.1) | 45.8 (-0.6) | 33.0 (-1.1) | **72.6** (+0.6) |

Table 4: The ablation experiments demonstrate the full roadmap from PVT-Tiny to TransNeXt-Micro. In step 1, PVTv2 introduces Overlapping Patch Embedding and Convolutional Feed-Forward (ConvFFN). In step 2, we made PVTv2 consistent with TransNeXt-Tiny in terms of height and width, with a head dimension of 48. In step 3, we decreased the head dimension to 24 and increased the number of attention heads.

Table 4 presents a comprehensive roadmap for upgrading PVT-Tiny to TransNeXt-Micro. To evaluate the robustness of the performance models at each stage, we conducted experiments on ImageNet-1K (Deng et al., 2009), ImageNet-C (Hendrycks & Dietterich, 2019), ImageNet-A (Hendrycks et al., 2021), ImageNet-R (Hendrycks & Dietterich, 2019), ImageNet-Sketch (Wang et al., 2019), and ImageNet-V2 (Recht et al., 2019).

**Effectiveness of our method**: The efficacy of our proposed convolutional GLU (ConvGLU) , pixel-focused attention, positional attention, and query embedding is demonstrated through ablation experiments from step 4 to 8. In the stages of step 4 to 5, step 6, and step 7 to 8, we replaced convolutional feed-forward (ConvFFN) with ConvGLU, spatial-reduction attention (SRA) with pixel-focused attention (PFA), and pixel-focused attention with aggregated attention, respectively. These three substitutions resulted in accuracy improvements of 0.8%, 0.9%, and 0.7% on ImageNet-1K, and 4.3%, 3.4%, and 3.0% on the ImageNet-A test set, respectively, indicating the significant contribution of these three components to performance. Moreover, in step 4, replacing ConvFFN with GLU led to a significant performance decline, underscoring the necessity of the $3 \times 3$ depthwise convolution (Chollet, 2017) as conditional position encodings (CPE) (Chu et al., 2021b), particularly as PVTv2's SRA (Wang et al., 2021b) did not use any other positional encoding at this stage.

**Impact of model structure**: We adjusted the width and depth of PVTv2 and the number of attention heads to match those of TransNeXt-Micro in steps 1 to 3 to avoid the impact of model structure. During this period, we observed that a deeper and thinner model significantly enhances performance.

Reducing the head dimension from 48 to 24 resulted in only a 0.04% performance change, indicating that the performance gain from increasing attention heads is extremely limited.

**Understanding of query embedding**: The query embedding exhibits very unique properties. Incorporating query embedding effectively improved the performance on ImageNet-1K val and ImageNet-V2 test sets but somewhat reduced performance on ImageNet-A, ImageNet-R, ImageNet-Sketch test sets; its impact on ImageNet-C was very weak. Notably, ImageNet-1K val, ImageNet-V2, and ImageNet-C (a distorted test set of ImageNet-1K val) adopted the same sampling strategy as the ImageNet-1K training set, while ImageNet-A, ImagetNet-R, and ImageNet-Sketch did not follow this principle. We believe these experimental results reflect that query embedding restricts the model's response range to enhance current task performance rather than affecting generalization to all types of data. During the learning process, the model optimizes this learnable query token, implicitly learning what the optimal question for the current task is in each attention layer (from a Visual Question Answering (VQA) perspective). This perspective can well explain why in these out-of-distribution test sets, query embedding has a very weak impact on the performance of the ImageNet-C test set which uses the same sampling strategy as the training set. Therefore, we believe there is a potential trade-off here. In the case of TransNeXt, even with query embedding, our model still achieved state-of-the-art model robustness.

## B.2 MULTI-SCALE INFERENCE

| Model | Method | Inference Size | | | | | | |
|-------|--------|------|------|------|------|------|------|------|
| | | $224^2$ | $256^2$ | $320^2$ | $384^2$ | $480^2$ | $512^2$ | $640^2$ |
| TransNeXt-Tiny | **Normal Mode** | 84.0 | 84.3 | 84.3 | 84.6 | 83.8 | 83.2 | **81.6** |
| | No Length-scaling | 84.0 | 84.3 | 84.4 | 84.7 | 83.7 | 83.2 | 80.9 |
| | Interpolate RPE | 84.0 | 84.1 | 84.2 | 84.3 | 83.1 | 82.4 | 79.5 |
| | **Linear Mode** | 84.0 | 84.0 | 83.9 | 84.1 | 83.0 | 82.6 | 80.7 |
| RepLKNet-31B (Ding et al., 2022) | | 83.5 | 83.6 | 81.0 | 70.0 | 21.4 | 10.1 | 0.9 |
| SLaK-S (Liu et al., 2023) | | 83.8 | 83.8 | 83.2 | 79.6 | 65.7 | 63.7 | 61.4 |
| ConvNeXt-B (Liu et al., 2022b) | | 83.8 | 84.2 | 84.0 | 83.6 | 81.6 | 80.7 | 77.3 |
| TransNeXt-Mirco | Normal Mode | 82.5 | 82.8 | 82.9 | 83.1 | 82.1 | 81.6 | 79.3 |
| | Linear Mode | 82.5 | 82.5 | 82.4 | 82.3 | 80.9 | 80.3 | 77.6 |
| TransNeXt-Small | Normal Mode | 84.7 | 84.9 | 84.9 | 85.0 | 84.1 | 83.8 | 82.2 |
| | Linear Mode | 84.7 | 84.7 | 84.7 | 84.9 | 84.0 | 83.6 | 81.7 |
| TransNeXt-Base | Normal Mode | 84.8 | 85.1 | 85.1 | 85.5 | 84.7 | 84.3 | 82.8 |
| | Linear Mode | 84.8 | 85.0 | 84.9 | 85.1 | 84.1 | 83.5 | 81.5 |

Table 5: The table shows the top-1 accuracy of ImageNet-1K of $224^2$-size trained TransNeXt under **normal** and **linear** inference modes on multiple image input sizes. At the same time, the effects of length-scaled cosine attention and log-CPB on multi-scale inference were tested, and the pure convolution model was included for comparison.

**Linear complexity mode for inference**: We observe that in Equations 15 and 16, if we consistently set $H_p$ and $W_p$ as fixed values independent of the input size, the computational complexity of both pixel-focused attention and aggregated attention grows linearly with the length of the input sequence. In this scenario, both pixel-focused attention and aggregated attention can operate under a linear complexity mode. This linear mode endows TransNeXt with a computational complexity growth curve close to that of a pure convolutional network when inferring large-size images. We test the performance changes of $224^2$-size trained TransNeXt and two prevalent pure convolutional models at multiple resolutions. In the default normal mode, $H_p$ and $W_p$ of aggregated attention are $\frac{1}{32}$ of the input image size, while in the linear mode, $H_p$ and $W_p$ are fixed at $\frac{1}{32}$ of the training image size, *i.e.*, $7 \times 7$.

**Results and analysis**: As shown in Table 5 and Fig 6, our TransNeXt-Tiny achieves better multi-scale extrapolation performance than pure convolutional models in both normal and linear modes. At the

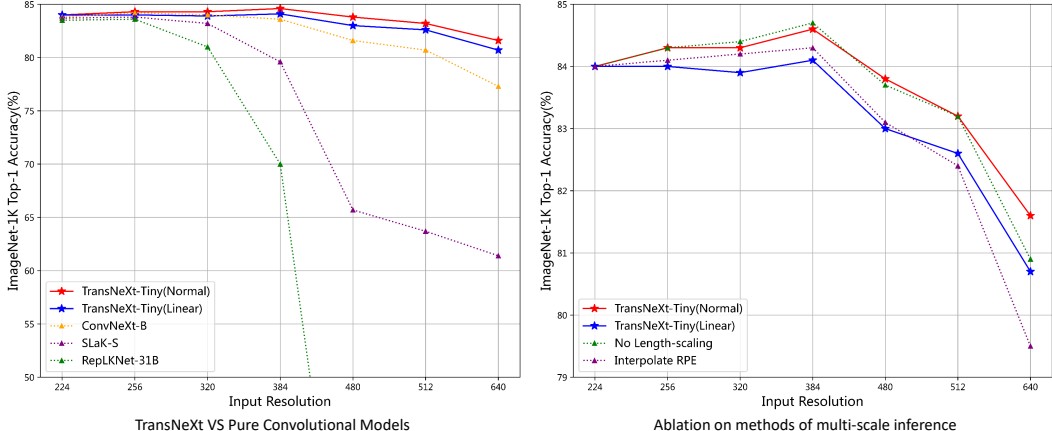

Figure 6: The left figure shows the comparison results of TransNeXt-Tiny under normal and linear inference modes with the pure convolution models on multi-scale image inference performance, while the right figure shows the impact of our position encoding design and length-scaled cosine attention on multi-scale image inference.

maximum resolution of $640^2$, the linear mode produces a performance decay of 0.5% to 1.7% relative to the normal mode, but such a trade-off still has advantages over pure convolutional models. As the image size increases, the performance decay of ConvNeXt-B is greater than that of TransNeXt's linear mode. RepLKNet-31B shows a more exaggerated performance decay, with a top-1 accuracy of only 0.9% at a resolution of $640^2$, which to some extent reveals the limitations of the super-large convolution kernel scheme. In traditional opinions, pure convolutional models have better multi-scale applicability than ViT models, and such experimental results also imply that this opinion needs to be re-examined.

**Impact of length-scaled cosine attention**: We compare the performance of length-scaled cosine attention with regular scaled cosine attention during multi-scale inference. According to Fig 6, length-scaling begins to take effect when the resolution reaches $640^2$. This implies that when the sequence length variation in softmax exceeds $8\times$, longer sequence lengths begin to significantly reduce the confidence of scaled cosine attention.

**Extrapolation vs Interpolation for relative position bias**: When a TransNeXt model trained at a resolution of $224^2$ infers at other sizes, we default to using log-CPB (Liu et al., 2022a) to extrapolate the $B_{(i,j)\sim\sigma(X)}$ under new resolutions from spatial relative coordinates $\Delta_{(i,j)\sim\sigma(X)}$. However, generating $\Delta_{(i,j)\sim\sigma(X)}$ cannot achieve the same speed as model inference. This is not a major issue in general because when the model needs to continuously infer at one or several new sizes, we only need to pre-calculate these new $\Delta_{(i,j)\sim\sigma(X)}$ and cache them. However, when the new inference resolution of the model is unknown and needs to change instantly according to input size, we need to use traditional interpolation schemes for relative position bias to interpolate $B_{(i,j)\sim\sigma(X)}$. As depicted in Fig 6, the input resolution of $640^2$ results in a significant performance degradation due to interpolation for relative position bias, surpassing that of the linear mode. This underscores the efficacy of log-CPB in extrapolating position bias. In our evaluation of UperNet with multi-scale and flip augmentations (Table14), we present test results under both interpolation and extrapolation for a balanced comparison, highlighting the influence of different schemes on multi-scale performance.

### B.3 ABLATION ON POSITIONAL ENCODING

**Results and analysis**: We conducted ablation experiments on the design of the relative position bias used in the sliding window path and pooling feature path in aggregated attention, with results shown in Table 6. When we completely removed the relative position bias $B_{(i,j)}$ used in aggregated attention, the model's performance significantly decreased by 0.8%. This indicates that using depthwise convolution to capture positional information from zero-padding is insufficient to represent the positional relationships of global tokens. When we used log-CPB to calculate the relative position bias of the sliding window, it also resulted in a 0.3% performance decline. This suggests that due to different feature scales, the numerical meanings of spatial coordinates $\Delta_{(i,j)\sim\sigma(X)}$ in the pooling feature path and $\Delta_{(i,j)\sim\rho(i,j)}$ in sliding window path are not exactly the same, highlighting the

| Method | Params(M) | FLOPs(G) | Top-1(%) |
|---|---|---|---|
| Remove $B_{(i,j)}$ | 28.1 | 5.6 | 83.2 |
| Calculate $B_{(i,j)\sim\rho(i,j)}$ by **log-CPB**$(\Delta_{(i,j)\sim\rho(i,j)})$ | 28.2 | 5.7 | 83.7 |
| Replace $B_{(i,j)\sim\rho(i,j)}$ by $Q_{(i,j)}T$ | 28.2 | 5.7 | 83.4 |
| Replace **log-CPB**$(\Delta_{(i,j)\sim\sigma(X)})$ by learnable $B_{(i,j)\sim\sigma(X)}$ | 28.1 | 5.6 | **84.0** |
| TransNeXt-Tiny | 28.2 | 5.7 | **84.0** |

Table 6: Ablation experiments on the design of relative position biases.

importance of using different methods to learn relative position bias in the two paths. Another consideration is to use dynamic relative position bias $Q_{(i,j)}T$ calculated by positional attention to replace $B_{(i,j)\sim\rho(i,j)}$, but this resulted in a significant performance decline of 0.6%. We believe this is due to inconsistencies in the behavior of the sliding window path and pooling path. The **log-CPB**$(\Delta_{(i,j)\sim\sigma(X)})$ calculated in the pooling path is static, while $Q_{(i,j)}T$ dynamically changes with input, and the two paths are coupled in the same softmax, causing interference with the mechanism of QKV attention. If we also use a learnable relative position bias $B_{(i,j)\sim\sigma(X)}$ instead of calculating by **log-CPB**$(\Delta_{(i,j)\sim\sigma(X)})$ in the pooling path, it does not affect model performance, but it does cause the model to lose its ability to extrapolate position biases for unknown size inputs. This demonstrates the similarity between the relative position biases calculated through log-CPB and those directly learned, also indicating that the log-CPB module is not the source of TransNeXt's high performance.

### B.4 Ablation on the Design of Convolutional GLU

We conducted ablation experiments on the design of convolutional GLU on the CIFAR-100 dataset using a 2M-sized model. We designed three optional variants, all using GELU as the activation function:

$$\textbf{ConvGLU}(X) = (XW_1 + B_1) \odot \text{GELU}(\text{DWConv}(XW_2 + B_2)) \tag{17}$$

$$\textbf{Type-1}(X) = (XW_1 + B_1) \odot \text{DWConv}(\text{GELU}(XW_2 + B_2)) \tag{18}$$

$$\textbf{Type-2}(X) = \text{DWConv}(XW_1 + B_1) \odot \text{GELU}(XW_2 + B_2) \tag{19}$$

$$\textbf{Type-3}(X) = \text{DWConv}((XW_1 + B_1) \odot \text{GELU}(XW_2 + B_2)) \tag{20}$$

The experiments, presented in Table 7, showed that our convolutional GLU, which follows the design philosophy of gated channel attention, is the optimal design. In Type-1, placing DWConv after the gated activation function disrupts the effect of setting value to zero in the gating branch. In Type-2, moving DWConv to the value branch causes a significant performance drop of 0.7% when a gating branch with a smaller receptive field controls a value branch with a larger receptive field, indicating that it is more reasonable to make gating decisions using a branch with a larger receptive field. In Type-3, adding a DWConv after the element-wise dot product result in the GLU module leads to the worst performance, suggesting that merely adding a DWConv to enhance local perceptual ability is not key to improving model performance with convolutional GLU.

### B.5 Ablation on Window Size

We conducted fast ablation experiments on CIFAR-100 (Krizhevsky & Hinton, 2009) using a 2M-sized model, results are reported in Table 8. Our observations indicate that an increase in the window size does not necessarily lead to an enhancement in the model's performance. We believe that these experimental results are due to the introduction of pooling features provides coarse-grained global perception abilities, greatly reducing the demand for single queries to perceive the sliding window field. Moreover, the fine-grained tokens overlap with the coarse-grained tokens, leading to additional inductive bias. Since the similarity results between queries and both fine-grained and coarse-grained

| Design | Params. (M) | FLOPs (G) | Top-1(%) |
|---|---|---|---|
| ConvGLU | 2.3 | 0.5 | **82.9** |
| Type-1 | 2.3 | 0.5 | 82.6 |
| Type-2 | 2.3 | 0.5 | 82.2 |
| Type-3 | 2.3 | 0.5 | 82.1 |

Table 7: Ablation study on the design of Convolutional GLU on CIFAR-100 (Krizhevsky & Hinton, 2009) dataset.

tokens compete in the same softmax, this approach benefits information aggregation in overlapping regions. However, as the window size increases, this inductive bias may not always be beneficial.

| Window Size | Params. (M) | FLOPs (G) | Top-1(%) |
|---|---|---|---|
| $3 \times 3$ | 2.3 | 0.50 | 82.9 |
| $5 \times 5$ | 2.3 | 0.52 | 82.0 |
| $7 \times 7$ | 2.3 | 0.54 | 82.9 |
| $9 \times 9$ | 2.3 | 0.57 | 82.5 |

Table 8: The ablation results of window size. We utilized a 2M-sized TransNeXt model to conduct experiments on the CIFAR-100 (Krizhevsky & Hinton, 2009) dataset under various window size settings.

### B.6 ABLATION ON MODEL ARCHITECTURE

To further explore the impact of model architecture on performance, we conducted ablation experiments based on TransNeXt-Micro. We attempted to replace aggregated attention with multi-head self-attention in stages 1-3 to observe its impact on model performance. The experimental results are presented in Table 9. We observed that when we replaced aggregated attention with multi-head self-attention in stage 3, where the number of blocks is the highest, the model performance decreased by 0.5%. Further replacement in stage 2 led to an additional 0.1% decline in performance. This suggests that our aggregated attention information aggregation method has advantages over global self-attention. When we tried to replace aggregated attention in stage 1, the model encountered an out-of-memory error on $8\times$ A100s with 80GB of memory, making it impossible to train the model with this configuration.

Under a resolution of $224^2$, $7 \times 7$ is the smallest size that can be achieved by integer multiple downsampling. For this reason, and to maintain consistency with PVTv2, our model opted for a pooling size of $\frac{1}{32}$ at each stage. However, in stage 4, the input resolution has already been reduced to $\frac{1}{32}$, rendering the downsampling module of aggregated attention ineffective. If aggregated attention is forcibly applied at this stage, features in the sliding window would be input into softmax twice through the pooling path, leading to distortion in importance calculation. Consequently, we selected MHSA in stage 4. At larger resolutions, such as $256^2$, we can employ a pooling size of $\frac{1}{64}$ at each stage to implement a model that fully utilizes aggregated attention at all stages. As demonstrated in Table 9, a micro-sized model that fully employs aggregated attention achieved an ImageNet-1K accuracy of 82.6% at a resolution of $256^2$.

### B.7 CUDA IMPLEMENTATION

In the native PyTorch (Paszke et al., 2019) implementation, feature extraction in the sliding window path is achieved through the unfold operation. The unfold operation involves two stages: 1) extracting the tensor within the sliding window through index access, and 2) explicitly creating a large tensor copy for the extracted tensor. This explicit feature extraction operation generates a huge temporary tensor and induces memory access pressure, which significantly reduces the model's speed. To address this, we introduce a CUDA operator implementation for calculating QK similarity and

| Token mixer | Input size | Window size | Pool size | Params. (M) | FLOPs (G) | Top-1(%) |
|---|---|---|---|---|---|---|
| **A-A-A-M** | $224^2$ | $3 \times 3$ | $7 \times 7$ | 12.8 | 2.7 | **82.5** |
| **A-A-M-M** | $224^2$ | $3 \times 3$ | $7 \times 7$ | 12.2 | 2.7 | 82.0 |
| **A-M-M-M** | $224^2$ | $3 \times 3$ | $7 \times 7$ | 12.2 | 2.9 | 81.9 |
| **M-M-M-M** | $224^2$ | $3 \times 3$ | $7 \times 7$ | 12.2 | 4.7 | **OOM** |
| **A-A-A-A** | $256^2$ | $3 \times 3$ | $4 \times 4$ | 13.1 | 3.3 | **82.6** |

Table 9: Ablation study on model architecture on ImageNet-1K dataset. **OOM** means out of memory error.

| Model | Throughput of inference | | | Duration of training (sec/iter) | | | Memory usage (GB) | | |
|---|---|---|---|---|---|---|---|---|---|
| | CUDA | Pytorch | Acceleration | CUDA | Pytorch | Acceleration | CUDA | Pytorch | Saving |
| TransNeXt-Micro | 1117 | 701 | +59.3% | 0.218 | 0.401 | +83.9% | 14.8 | 17.8 | 16.8% |
| TransNeXt-Tiny | 756 | 471 | +60.5% | 0.315 | 0.609 | +93.3% | 23.2 | 27.3 | 15.0% |
| TransNeXt-Small | 394 | 246 | +60.2% | 0.595 | 1.161 | +95.1% | 41.6 | 49.3 | 15.6% |
| TransNeXt-Base | 297 | 186 | +59.6% | 0.771 | 1.568 | +103.4% | 58.1 | 68.6 | 15.3% |

Table 10: Performance comparison between CUDA implementation and native PyTorch implementation. We measure throughput using a batch size of 64 on a single V100 with 16GB of memory under FP16, while the iteration time and memory consumption during training are measured on $8\times$ A100s (PCIe) with a total batch size of 1024 under automatic mixed precision.

aggregating value by attention weights in the sliding window path. This implementation circumvents the need for explicit tensor extraction from the sliding window, thereby markedly enhancing the model's throughput and training speed. As shown in Table 10, our CUDA implementation provides up to 60.5% acceleration for inference, up to 103.4% acceleration for training and saves up to 16.8% of memory consumption for training.

## C  DETAILED FEATURE ANALYSIS

### C.1  TRANSLATIONAL EQUIVARIANCE

Both our pixel-focused attention and aggregated attention employ two types of tokens that interact with each query: those centered at the present query and those shared among all queries on the feature map. Notably, both of these attention mechanisms only uses relative position bias without absolute position encoding, resulting in translational equivariance (Wennberg & Henter, 2021; Shaw et al., 2018).

### C.2  COMPARISON WITH PRIOR WORK

Table 11 illustrates the differences in the information aggregation methods between our pixel-focused attention, aggregated attention, and the token mixers from previous works. Within the realm of attention mechanisms, relative position bias (Shaw et al., 2018) can play a role similar to depthwise convolution (Chollet, 2017), using a static affinity matrix for information aggregation. From this perspective, our aggregated attention achieves a unified aggregation of the static affinity matrix, QKV attention, LKV attention, and QLV attention. This makes it the most diversified token mixer in terms of information aggregation methods to date.

## D  DOWNSTREAM EXPERIMENTAL RESULTS

## E  VISUALIZATION BASED ON EFFECTIVE RECEPTIVE FIELD

We employ the Effective Receptive Field (ERF) (Luo et al., 2016) method as a visualization tool to analyze the information aggregation approach of TransNeXt. In Fig 7, we visualize the ERF of

| Method | Focus Prior | Global Perception | Translational Equivariance | Static Affinity Matrix | Dynamic Affinity Matrix (QKV) | Dynamic Affinity Matrix (LKV) | Dynamic Affinity Matrix (QLV) |
|---|---|---|---|---|---|---|---|
| CNNs (LeCun et al., 1995) | ✓ | | ✓ | ✓ | | | |
| MLP-Mixer (Tolstikhin et al., 2021) | | ✓ | | ✓ | | | |
| ViT(APE) (Dosovitskiy et al., 2021) | | ✓ | | | ✓ | | |
| ViT(RPB) (Dosovitskiy et al., 2021) | | ✓ | ✓ | ✓ | ✓ | | |
| PVT (Wang et al., 2021a) | | ✓ | ✓ | | ✓ | | |
| Focal-Transformer (Yang et al., 2021) | ✓ | ✓ | | ✓ | ✓ | | |
| FocalNet (Yang et al., 2022) | ✓ | ✓ | ✓ | ✓ | | | |
| VOLO (Yuan et al., 2023) | ✓ | | ✓ | | | | ✓ |
| Involution (Li et al., 2021) | ✓ | | ✓ | | | | ✓ |
| QnA (Arar et al., 2022) | ✓ | | ✓ | | | ✓ | |
| **Pixel-focused Attention (Ours)** | ✓ | ✓ | ✓ | ✓ | ✓ | | |
| **Aggregated Attention (Ours)** | ✓ | ✓ | ✓ | ✓ | ✓ | ✓ | ✓ |

Table 11: Comparison of our method with the information aggregation approach of the token mixer in prior work. Here **APE=A**bsolute **P**ositional **E**ncoding, and **RPB=R**elative **P**osition **B**ias.

| Backbone | Encoder size(M) | #Params. (M) | $AP^b$ | $AP_{50}^b$ | $AP_{75}^b$ | $AP^m$ | $AP_{50}^m$ | $AP_{75}^m$ |
|---|---|---|---|---|---|---|---|---|
| Swin-T (Liu et al., 2021) | 28.3 | 47.8 | 43.7 | 66.6 | 47.7 | 39.8 | 63.3 | 42.7 |
| PVTv2-B2 (Wang et al., 2021b) | 25.4 | 45.3 | 45.3 | 67.1 | 49.6 | 41.2 | 64.2 | 44.4 |
| FocalNet-T (LRF) (Yang et al., 2022) | 28.6 | 48.9 | 46.1 | 68.2 | 50.6 | 41.5 | 65.1 | 44.5 |
| Swin-S (Liu et al., 2021) | 49.6 | 69.1 | 46.5 | 68.7 | 51.3 | 42.1 | 65.8 | 45.2 |
| CSWin-T (Dong et al., 2022) | 23 | 42 | 46.7 | 68.6 | 51.3 | 42.2 | 65.6 | 45.4 |
| Swin-B (Liu et al., 2021) | 87.8 | 107.1 | 46.9 | 69.2 | 51.6 | 42.3 | 66.0 | 45.5 |
| PVTv2-B3 (Wang et al., 2021b) | 45.2 | 64.9 | 47.0 | 68.1 | 51.7 | 42.5 | 65.7 | 45.7 |
| InternImage-T (Wang et al., 2023) | 30 | 49 | 47.2 | 69.0 | 52.1 | 42.5 | 66.1 | 45.8 |
| PVTv2-B5 (Wang et al., 2021b) | 82.0 | 101.6 | 47.4 | 68.6 | 51.9 | 42.5 | 65.7 | 46.0 |
| PVTv2-B4 (Wang et al., 2021b) | 62.6 | 82.2 | 47.5 | 68.7 | 52.0 | 42.7 | 66.1 | 46.1 |
| InternImage-S (Wang et al., 2023) | 50 | 69 | 47.8 | 69.8 | 52.8 | 43.3 | 67.1 | 46.7 |
| SMT-S (Lin et al., 2023) | 20.5 | 40.0 | 47.8 | 69.5 | 52.1 | 43.0 | 66.6 | 46.1 |
| BiFormer-S (Zhu et al., 2023) | 25.5 | 45.2 | 47.8 | 69.8 | 52.3 | 43.2 | 66.8 | 46.5 |
| CSWin-S (Dong et al., 2022) | 35 | 54 | 47.9 | 70.1 | 52.6 | 43.2 | 67.1 | 46.2 |
| FocalNet-S (LRF) (Yang et al., 2022) | 50.3 | 72.3 | 48.3 | 70.5 | 53.1 | 43.1 | 67.4 | 46.2 |
| BiFormer-B (Zhu et al., 2023) | 56.8 | 76.3 | 48.6 | 70.5 | 53.8 | 43.7 | 67.6 | 47.1 |
| CSWin-B (Dong et al., 2022) | 78 | 97 | 48.7 | 70.4 | 53.9 | 43.9 | 67.8 | 47.3 |
| InternImage-B (Wang et al., 2023) | 97 | 115 | 48.8 | 70.9 | 54.0 | 44.0 | 67.8 | 47.4 |
| SMT-B (Lin et al., 2023) | 32 | 51.7 | 49.0 | 70.2 | 53.7 | 44.0 | 67.6 | 47.4 |
| FocalNet-B (LRF) (Yang et al., 2022) | 88.7 | 111.4 | 49.0 | 70.9 | 53.9 | 43.5 | 67.9 | 46.7 |
| **TransNeXt-Tiny (Ours)** | 28.2 | 47.9 | **49.9** | 70.5 | 53.7 | 43.9 | 67.4 | **47.5** |
| **TransNeXt-Small (Ours)** | 49.7 | 69.3 | **51.1** | 72.6 | 56.2 | **45.5** | 69.8 | 49.1 |
| **TransNeXt-Base (Ours)** | 89.7 | 109.2 | **51.7** | 73.2 | 56.9 | **45.9** | 70.5 | 49.7 |

Table 12: Detailed COCO object detection and instance segmentation results using the Mask R-CNN (He et al., 2020) $1\times$ schedule, sorted in ascending order based on $AP^b$ performance..

| Model | Encoder size(M) | #Params. (M) | Epochs | scales | Pre-trained | $AP^b$ |
|---|---|---|---|---|---|---|
| ConvNeXt-B (Liu et al., 2022b) | 88.6 | 110 | 12 | 4 | IN-1K ($384^2$) | 52.6 |
| ConvNeXt-L (Liu et al., 2022b) | 198 | 221 | 12 | 4 | IN-1K ($384^2$) | 53.4 |
| **TransNeXt-Tiny (Ours)** | **28.2** | **47.8** | 12 | 4 | IN-1K (**$224^2$**) | **55.1** |
| **TransNeXt-Tiny (Ours)** | 28.2 | 48.1 | 12 | 5 | IN-1K ($224^2$) | **55.7** |
| **TransNeXt-Small (Ours)** | 49.7 | 69.6 | 12 | 5 | IN-1K ($224^2$) | **56.6** |
| **TransNeXt-Base (Ours)** | 89.7 | 110 | 12 | 5 | IN-1K ($224^2$) | **57.1** |
| Swin-L (Liu et al., 2021) | 197 | 218 | 12 | 5 | IN-22K ($384^2$) | 57.2 |

Table 13: Comparison of object detection results on the COCO dataset using the DINO method. The results are sorted in ascending order based on the $AP^b$ scores.

| Model | Encoder size(M) | #Params. (M) | Crop -size | Pre-trained | mIoU (%) | +MS (%) |
|---|---|---|---|---|---|---|
| Swin-T (Liu et al., 2021) | 28.3 | 60 | $512^2$ | IN-1K | 44.5 | 45.8 |
| Focal-T (Yang et al., 2021) | 29.1 | 62 | $512^2$ | IN-1K | 45.8 | 47.0 |
| ConvNeXt-T (Liu et al., 2022b) | 28.6 | 60 | $512^2$ | IN-1K | 46.0 | 46.7 |
| FocalNet-T(LRF) (Yang et al., 2022) | 28.6 | 61 | $512^2$ | IN-1K | 46.8 | 47.8 |
| Swin-S (Liu et al., 2021) | 49.6 | 81 | $512^2$ | IN-1K | 47.6 | 49.5 |
| UniFormer-S (Li et al., 2022b) | 22 | 52 | $512^2$ | IN-1K | 47.6 | 48.5 |
| Focal-S (Yang et al., 2021) | 51.1 | 85 | $512^2$ | IN-1K | 48.0 | 50.0 |
| Swin-B (Liu et al., 2021) | 87.8 | 121 | $512^2$ | IN-1K | 48.1 | 49.7 |
| ConvNeXt-S (Liu et al., 2022b) | 50.2 | 82 | $512^2$ | IN-1K | 48.7 | 49.6 |
| Focal-B (Yang et al., 2021) | 89.8 | 126 | $512^2$ | IN-1K | 49.0 | 50.5 |
| FocalNet-S(LRF) (Yang et al., 2022) | 50.3 | 84 | $512^2$ | IN-1K | 49.1 | 50.1 |
| ConvNeXt-B (Liu et al., 2022b) | 88.6 | 122 | $512^2$ | IN-1K | 49.1 | 49.9 |
| SMT-S (Lin et al., 2023) | 20.5 | 50.1 | $512^2$ | IN-1K | 49.2 | 50.2 |
| SMT-B (Lin et al., 2023) | 32 | 61.8 | $512^2$ | IN-1K | 49.6 | 50.6 |
| UniFormer-B (Li et al., 2022b) | 49.8 | 80 | $512^2$ | IN-1K | 50.0 | 50.8 |
| FocalNet-B(LRF) (Yang et al., 2022) | 88.7 | 126 | $512^2$ | IN-1K | 50.5 | 51.4 |
| **TransNeXt-Tiny (Ours)** | **28.2** | **59** | $512^2$ | **IN-1K** | **51.1** | **51.5/51.7** |
| **TransNeXt-Small (Ours)** | **49.7** | **80** | $512^2$ | **IN-1K** | **52.2** | **52.5/52.8** |
| ConvNeXt-B (Liu et al., 2022b) | 88.6 | 122 | $640^2$ | IN-22K | 52.6 | 53.1 |
| **TransNeXt-Base (Ours)** | **89.7** | **121** | $\mathbf{512^2}$ | **IN-1K** | **53.0** | **53.5/53.7** |

Table 14: A comprehensive comparison of semantic segmentation results on the ADE20K dataset using the UperNet method. +MS for evaluation with multi-scale and flip augmentations. In the context of multi-scale evaluation, TransNeXt reports test results under two distinct scenarios: interpolation and extrapolation of relative position bias. The results are sorted in ascending order based on the mIoU scores.

| Model | Encoder size(M) | #Params. (M) | Crop -size | Pre-trained | mIoU(%) |
|---|---|---|---|---|---|
| Swin-S (Liu et al., 2021) | 49.6 | 68.8 | $512^2$ | IN-1K ($224^2$) | 51.2 |
| Swin-B (Liu et al., 2021) | 87.8 | 107 | $640^2$ | IN-1K ($224^2$) | 52.4 |
| **TransNeXt-Tiny (Ours)** | **28.2** | **47.5** | $\mathbf{512^2}$ | **IN-1K ($224^2$)** | **53.4** |
| Swin-B (Liu et al., 2021) | 87.8 | 107 | $640^2$ | IN-22K ($384^2$) | 53.9 |
| **TransNeXt-Small (Ours)** | **49.7** | **69.0** | $\mathbf{512^2}$ | **IN-1K ($224^2$)** | **54.1** |
| **TransNeXt-Base (Ours)** | **89.7** | **109** | $\mathbf{512^2}$ | **IN-1K ($224^2$)** | **54.7** |

Table 15: Comparison of semantic segmentation results on the ADE20K dataset using the Mask2Former method. The results are sorted in ascending order based on the mIoU scores.

four encoder stages for TransNeXt-Tiny, ConvNeXt-T, and Swin-T. In Fig 8, we further conduct a comprehensive ERF visualization comparison on the fourth stage of the models on ImageNet-A, ImageNet-Sketch, and ImageNet-C datasets.

Our observations are as follows:

1. In the ERF visualization, TransNeXt-Tiny outperforms Swin-T, CSwin-T, and ConvNeXt-T by Stage 3, demonstrating a more natural and smoother ERF. In contrast, ConvNeXt, Swin, and CSwin exhibit distinct blocky patterns, which we attribute to artifacts from their token mixer designs. Despite the presence of multiple layers, these token mixers are unable to eliminate artifacts induced by window-based local attention or convolution kernels, resulting in an unnatural information mixing. This observation supports the experimental evidence that deep networks with residual blocks function as ensembles of shallower networks, highlighting the significance of a single token mixer in achieving a local-global modeling approach that is more akin to biological vision. TransNeXt's ERF represents a method of information perception that is closer to biological vision, achieving a natural visual focus and validating its biomimetic design's effectiveness.

2. In a comprehensive visualization evaluation across multiple out-of-distribution test sets, TransNeXt-Tiny demonstrates a more adaptive information perception method. Its effective receptive field's information perception method undergoes significant changes with different datasets. This change can be clearly observed at multiple severity levels on ImageNet-C. Meanwhile, Swin-T's ERF exhibits a similar pattern across all test sets, and ConvNeXt-T's ERF lies somewhere in between. We believe that a more adaptive ERF reflects the model's robustness, and such visualization comparison results are consistent with the robustness evaluation results in Table 1.

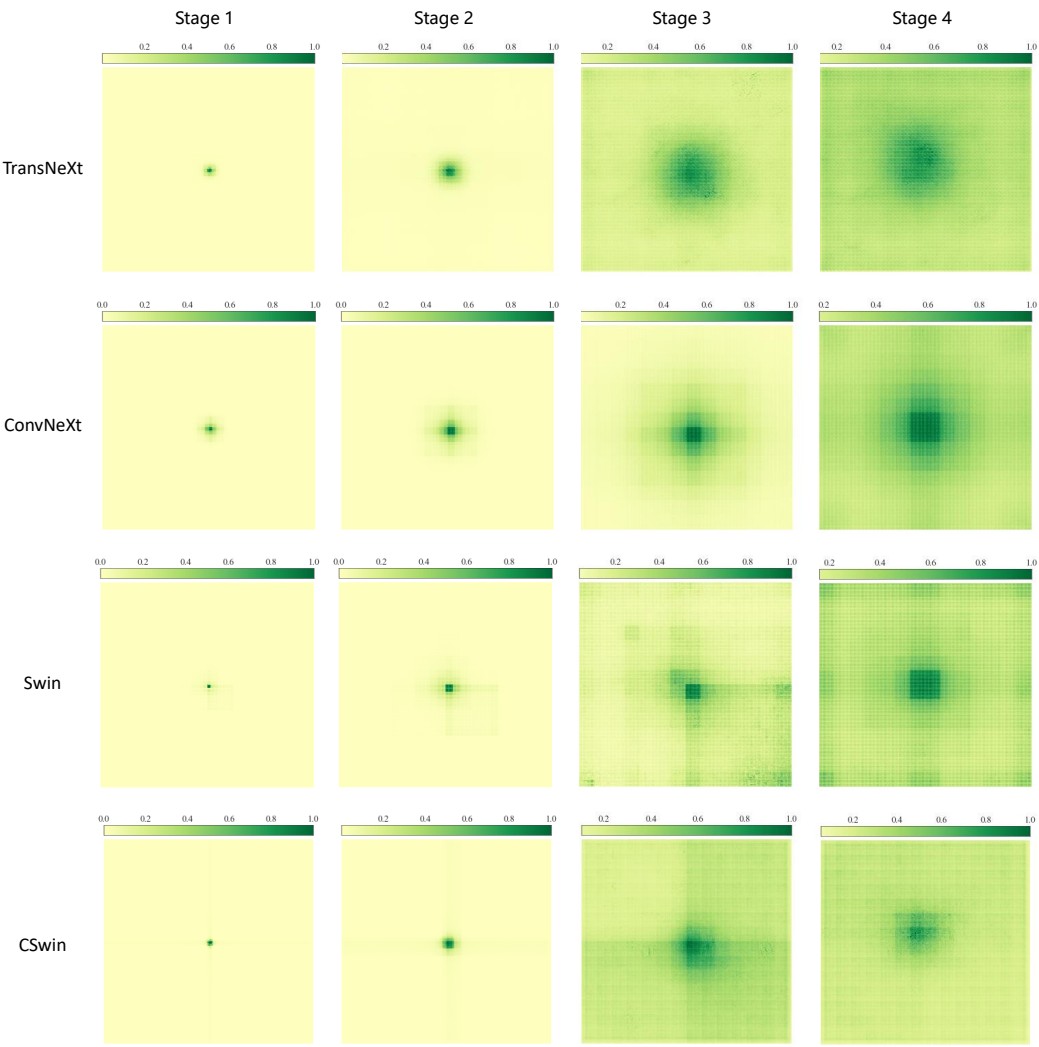

Figure 7: Visualization of the Effective Receptive Field (ERF) on ImageNet-1K validation set. Each visualization is based on an average of 5000 images with a resolution of $224 \times 224$. We visualize the ERFs of four stages for TransNeXt-Tiny, ConvNeXt-T, Swin-T and Cswin-T.

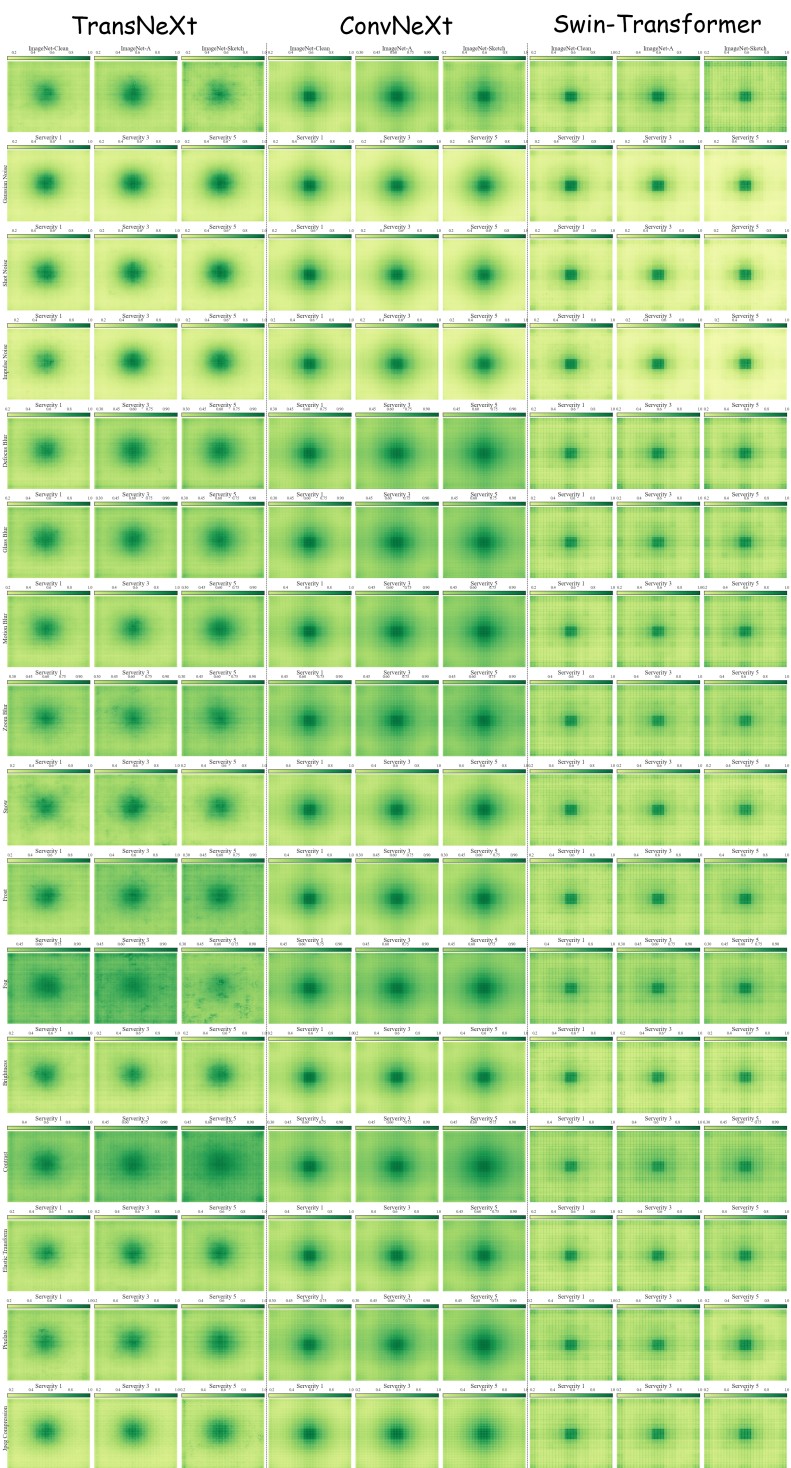

Figure 8: Visualization of the Effective Receptive Field (ERF) for TransNeXt-Tiny, ConvNeXt-T, and Swin-T on various datasets including ImageNet-1K validation set (Clean), ImageNet-Adversarial, ImageNet-Sketch, and ImageNet-C. The visual analysis diagrams for ImageNet-C commence from the second row of the figure. For each corruption mode, we have included visual images with severity levels of 1, 3, and 5. Each ERF image is generated by averaging over 5000 images with a resolution of $224 \times 224$ from each dataset.

