# OpenReview forum: "TransNeXt: Aggregating Diverse Attentions in One Vision Model"
_ICLR.cc/2024/Conference — ICLR 2024 Conference Withdrawn Submission_

### Official Review · Reviewer_2Wbt · 2023-10-28

**Soundness:** 3 good
**Presentation:** 3 good
**Contribution:** 3 good
**Rating:** 5
**Confidence:** 5

**Summary:**

This paper presents TransNeXt, a novel biomimetic design-based token mixer that combines fine-grained attention to neighboring tokens and coarse-grained attention to global features, taking into account spatial information aggregation. The proposed method incorporates learnable tokens that interact with conventional queries and keys, enabling the generation of affinity matrices that go beyond relying solely on the similarity between queries and keys. Additionally, the paper introduces Convolutional GLU, a channel mixer that bridges the gap between GLU and SE mechanisms, facilitating channel attention based on neighboring features.

To evaluate the effectiveness of the proposed method, experiments were conducted on various benchmark datasets. These include ImageNet for image classification tasks, COCO for object detection tasks, and ADE20K for semantic segmentation tasks. The results of these experiments demonstrate the efficacy of the proposed method in achieving state-of-the-art performance across these different tasks.

**Strengths:**

1. The paper is commendably well-written, effectively presenting its ideas in a clear and coherent manner. The proposed framework is explained in a manner that facilitates easy comprehension for readers.

2. The incorporation of pixel-focused attention, which encompasses both fine-grained local attention and coarse-grained global attention while engaging in competition, is a notable aspect of the research. Furthermore, the integration of query embedding and positional attention mechanisms within the pixel-focused attention framework enhances the generation of affinity matrices. This diversification of the affinity matrix generation process moves beyond a sole reliance on query-key similarity, enabling the aggregation of multiple attention mechanisms within a single attention layer.

3. An additional contribution of the paper is the introduction of Convolutional GLU as a novel channel mixer. This component bridges the gap between GLU and SE mechanisms, facilitating channel attention based on neighboring features. The incorporation of Convolutional GLU adds a valuable element to the proposed method.

4. The effectiveness of the proposed method is demonstrated through comprehensive experiments conducted on various benchmarks, including image classification, object detection, and semantic segmentation tasks. The obtained results showcase the state-of-the-art performance achieved across these diverse tasks, further validating the efficacy of the proposed approach.

**Weaknesses:**

1. It is important to note that the combination of local attention and global attention mechanisms, as well as the incorporation of competition between different grained attention, have been explored in previous works on architecture design. These mechanisms are not novel and have been utilized in the context of related research. It is crucial to acknowledge the existing literature and the contributions made by previous studies in these areas.

[1] Jiang et al.Dual Path Transformer with Partition Attention, in Arxiv 2023.

2. While providing the theoretical complexity of the proposed Aggregated Attention is informative, it is indeed crucial to consider latency comparisons as well. Latency does not always correlate directly with model parameters and FLOPs (floating-point operations per second). Therefore, it is essential to conduct comparisons with different methods on the same device to assess the real-world performance in terms of latency. This empirical evaluation would provide valuable insights into the practical efficiency of the proposed method and enable a more comprehensive assessment of its performance.

3. While this paper offers an in-depth examination of each element within the Aggregated Attention model, it appears to primarily combine existing technologies to enhance performance. Could you kindly summarize the three principal components of the proposed Aggregated Attention model?

**Questions:**

The running speed of the proposed method in terms of efficiency is an important aspect to consider. It would be valuable to compare the efficiency of the proposed method with related works to assess its performance in this regard. By conducting a comparative analysis, we can gain insights into how the proposed method fares in terms of running speed and efficiency when compared to existing approaches in the field.

---

> ### Author Response · Authors · 2023-11-13
> **Responses to Reviewer 4 #part.1**
>
> We express our sincere gratitude to the reviewers for their recognition of our paper as “commendably well-written”, acknowledging our PFA module as “a notable aspect”, and endorsing the Convolutional GLU as “a valuable element”. These affirmations serve as a significant motivation for our work, which has taken over a year to complete.
>
> **R1: Novelty and Distinction from Previous Work**
>
> We highly appreciate the reviewers’ insightful comment that the combination of global and local attention, as well as competition at different granularities, has been explored in previous work. This indicates that our previous description of our contributions did not adequately express the innovative aspects of our work compared to previous studies. We have revised these descriptions in the new version of our paper.
>
> Furthermore, we are grateful for the papers provided by the reviewers. Citing previous work is crucial, and we have referenced it in the revised version of our paper.
>
> We believe that the contribution and novelty of our work lie in proposing an attention mechanism that aligns more closely with biological foveal vision than previous work. Its pixel-wise operation can effectively simulate the continuous movement of the eyeball, while each pixel possesses a comprehensive global perception. (In contrast to our approach, Focal attention operates on a window-wise manner, and its method based on window partitioning also deviates from foveal vision of human eyes.)
>
> The motivation for proposing PFA is that previous work and experiments have found that deep networks with residual blocks behave like ensembles of shallower networks. This implies that cross-layer information exchange is not as effective as expected. In convolutional networks, this is manifested as the receptive field of deep networks based on stacked convolutional kernels not being as large as expected (hence the proposal of the super-large convolutional kernel scheme). So, what about ViT models? We found that in various efficient ViT models, the information exchange between cross-window or global and local features formed by stacking blocks is not as sufficient as expected. This can be clearly seen in our Appendix E based on Effective Receptive Field (ERF) visualization. Models based on local windows (Swin, CSwin) present unique blocky patterns in visual perception, and these pattern styles are closely related to their window design methods. This indicates that even after many layers of stacking, the exchange of information across windows is still insufficient, resulting in unnatural information mixing. Therefore, we believe that it is very important to implement a pixel-wise global-local perception mode that is closer to biological vision and can simulate eyeball movement in a single token mixer, so we proposed PFA and subsequent Aggregated Attention. The visualization based on ERF shows that our method forms a very natural and smooth information perception mode.
>
> In Appendix B.2, we conducted a comparison with the super-large convolutional kernel scheme. It can be seen that our model has more advantages in large-scale image inference than RepLKNet(CVPR 2022) and SLaK(ICLR 2023), while the latter two show some limitations in large-size image inference. We believe that this comparative experiment is enlightening for the subsequent research of the computer vision community in related fields.

---

> > ### Author Response · Authors · 2023-11-13
> > **Responses to Reviewer 4 #part.2**
> >
> > **R2 & Q1: Throughput and Latency Test**
> >
> > Currently, we have only implemented native CUDA code without extensive optimization. Consequently, it is not as efficient as highly optimized dense GPU operators. We tested the model’s throughput on a V100 16G with FP32 precision at a batch size of 64, and the model’s latency at a batch size of 10. Our biomimetic vision implementation based on the sliding window is more efficient than the Focal Transformer method. Moreover, compared to previous models such as MaxViT (ECCV 2022), QuadTree (ICLR 2022), and BiFormer (CVPR 2023), our model achieved competitive TOP-1 accuracy results under similar throughput conditions.
> >
> > | Models | Params | FLOPs（G） | TOP-1 | throughputs(img/s) | latency(ms) |
> > | --- | --- | --- | --- | --- | --- |
> > | BiFormer-T | 13.1 | 2.2 | 81.4 | 828 | 1.73 |
> > | Swin-T | 28.3 | 4.5 | 81.2 | 790 | 1.44 |
> > | ConvNeXt-T | 28.6 | 4.5 | 82.3 | 779 | 1.42 |
> > | QuadTree-B-b1 | 13.6 | 2.3 | 80.0 | 663 | 2.85 |
> > | **TransNeXt-Micro** | 12.8 | 2.7 | **82.5** | 641 | 3.95 |
> > | Swin-S | 49.6 | 8.7 | 83.1 | 460 | 2.51 |
> > | MaxViT-Tiny | 30.9 | 5.6 | 83.4 | 459 | 2.77 |
> > | ConvNeXt-S | 50.2 | 8.7 | 83.1 | 441 | 2.54 |
> > | **TransNeXt-Tiny** | 28.2 | 5.7 | **84.0** | 413 | 3.98 |
> > | BiFormer-S | 25.5 | 4.5 | 83.8 | 396 | 3.62 |
> > | QuadTree-B-b2 | 24.2 | 4.5 | 82.7 | 361 | 5.51 |
> > | Focal-Transformer-T | 29.1 | 4.9 | 82.2 | 337 | 3.3 |
> > | Swin-B | 87.8 | 15.4 | 83.5 | 292 | 3.73 |
> > | ConvNeXt-B | 88.6 | 15.4 | 83.8 | 290 | 3.8 |
> > | MaxViT-Small | 68.9 | 11.7 | 84.4 | 273 | 4.2 |
> > | BiFormer-B | 56.8 | 9.8 | 84.3 | 241 | 4.7 |
> > | QuadTree-B-b3 | 46.3 | 7.8 | 83.7 | 238 | 8.9 |
> > | **TransNeXt-Small** | 49.7 | 10.3 | **84.7** | 214 | 6.85 |
> > | Focal-Transformer-S | 51.1 | 9.1 | 83.5 | 203 | 6.46 |
> > | QuadTree-B-b4 | 64.2 | 11.5 | 84.0 | 166 | 13.17 |
> > | **TransNeXt-Base** | 89.7 | 18.4 | **84.8** | 151 | 7.38 |
> > | Focal-Transformer-B | 89.8 | 16 | 83.8 | 145 | 7.63 |
> > | MaxViT-Base | 119.5 | 24 | 84.9 | 144 | 8.13 |
> >
> > Furthermore, the speed of TransNeXt is expected to improve with more engineering efforts. We will continue to provide more efficient operator optimizations in the future to enhance the competitiveness of TransNeXt.

---

> ### Author Response · Authors · 2023-11-13
> **Responses to Reviewer 4 #part.3**
>
> **R3: Summary of Principal Components in the Model**
>
> Our manuscript and model encapsulate a series of innovative points accomplished over the span of more than a year. Our novel contributions primarily manifest in the following aspects:
>
> 1. A biomimetic attention mechanism, Pixel-focused attention, which is highly consistent with the biological foveal visual system and capable of simulating continuous eye movements.
> 2. A novel channel mixer, Convolutional GLU, which possesses channel attention based on nearest neighbor features.
> 3. Length-scaled cosine attention, which enhances the extrapolation of existing attention mechanisms for multi-scale inputs.
>
> In the review process, we found that each reviewer acknowledged the novelty and contribution of different components. Reviewer 1 recognized the novelty of the PFA module, Reviewer 3 acknowledged the contribution of Length-scaled cosine attention in multi-scale inference, and Reviewer 4 recognized the PFA module as "a notable aspect" and Convolutional GLU as "a valuable element". We express our sincere gratitude to the reviewers for their recognition of these contributions.
>
> The main concern raised in the review comments was that the introduction of QLV and LKV attention operations in PFA made our method appear to be a combination of existing technologies. We believe that such concerns are reasonable and insightful. This indicates that our paper did not effectively explain the advantages and competitiveness of our method of integrating these three mechanisms. We have revised the relevant statements in the latest revision. Our integration method requires very little additional overhead, is highly efficient, and has novelty in the way it is integrated.
>
> Since the introduction of the Non-QKV mechanism in the Synthesizer(ICML 2021) paper, works such as QnA (CVPR 2022) and VOLO(TPAMI 2023)/Involution(CVPR 2021) have validated the feasibility of LKV and QLV mechanisms in visual models. We note that no previous work has attempted to unify QKV, LKV, and QLV, these three attention mechanisms, in a single attention layer (we believe that merely implementing through block stacking or ensemble methods would be trivial and lack novelty, as these methods would also be affected by the potential depth degradation of residual connections). We consider this work to be the first attempt to unify these three attention mechanisms in a single attention layer.
>
> Our design of PFA serves as a promising foundation for unifying these three attention mechanisms. It inherently has a sliding window attention branch, and reusing this branch reduces the overhead required by the QLV mechanism to $HWk^2C$. Our method of introducing the LKV mechanism is more efficient, requiring only the addition of a query embedding to all queries to achieve the sum of the LK affinity matrix and the QK affinity matrix, with its additional overhead being negligible. In terms of experimental results, enhancing PFA to Aggregated Attention requires only about **0.2%(of base model) to 0.3%(of  micro model)** of the additional computational consumption in the entire model, but the improvement is significant, achieving a very cost-effective trade-off. From this perspective, our method of unifying QKV, QLV, and LKV attentions does not employ a trivial approach, and is both efficient and has its innovative aspects. We believe this is a successful attempt.
>
> | Models | Params | FLOPs（G） | IN-1K | IN-A |
> | --- | --- | --- | --- | --- |
> |TransNeXt-Micro(FPA)|12.78|2.65|81.8|26.9|
> |TransNeXt-Micro(AA)|12.81 (+0.2%)|2.66(+0.3%)|82.5(+0.7%)|29.9(+3.0%)|
>
> We once again express our gratitude to the reviewers for their meticulous review and insightful suggestions regarding our paper.

---

### Official Review · Reviewer_vXBQ · 2023-10-30

**Soundness:** 3 good
**Presentation:** 4 excellent
**Contribution:** 3 good
**Rating:** 5
**Confidence:** 5

**Summary:**

In this paper, the authors introduce modifications to the Vision Transformers (ViTs) design, chiefly, the Aggregated Attention mechanism and the Convolutional Gated Linear Unit (GLU). The Aggregated Attention is biomimetic and facilitates each token to attend to both nearest neighbor features and global features finely and coarsely, respectively. It harmoniously integrates multiple attention mechanisms within a single layer, eradicating the necessity for alternating stacking of various token mixers. This novel attention mechanism also imbues the model with the richness of pixel-focused attention and relative positional bias, improving the models' ability to aggregate essential spatial information and enhance their translational equivariance. The paper also proposes a Convolutional GLU, a novel channel mixer adept for image-related tasks, which bridges the gap between the conventional GLU and Squeeze-and-Excitation (SE) mechanisms. It leverages local feature-based channel attention, bolstering the model's robustness and local modeling capabilities. This architecture exhibits commendable performance, standing at the pinnacle across multiple model sizes.

**Strengths:**

The proposed Length-scaled cosine attention improves existing attention mechanisms by enhancing extrapolation capabilities, enabling models to adeptly manage and interpret multi-scale image inputs, fostering better adaptability and effectiveness in processing diverse image sizes and scales.

**Weaknesses:**

1. Novelty is limited as the method proposed, where queries attend to both fine-grained and coarse-grained information simultaneously, has been extensively studied previously [1,2].
2. More ablation studies are needed. In Table 4, step 5, the paper only reports performance gains of PFA over SRA. Since PFA is central to this paper, it would be insightful to see if replacing SRA with Cross-Shaped Window Self-Attention from CSWin[3], Focal Self-attention[1], or Shunted Attention[4] would yield higher performance.

[1] Yang, J., Li, C., Zhang, P., Dai, X., Xiao, B., Yuan, L., & Gao, J. (2021). Focal attention for long-range interactions in vision transformers. Advances in Neural Information Processing Systems, 34, 30008-30022.
[2] Chen, M., Lin, M., Li, K., Shen, Y., Wu, Y., Chao, F., & Ji, R. (2023, June). Cf-vit: A general coarse-to-fine method for vision transformer. In Proceedings of the AAAI Conference on Artificial Intelligence (Vol. 37, No. 6, pp. 7042-7052).
[3] Dong, X., Bao, J., Chen, D., Zhang, W., Yu, N., Yuan, L., ... & Guo, B. (2022). Cswin transformer: A general vision transformer backbone with cross-shaped windows. In Proceedings of the IEEE/CVF Conference on Computer Vision and Pattern Recognition (pp. 12124-12134).
[4] Ren, S., Zhou, D., He, S., Feng, J., & Wang, X. (2022). Shunted self-attention via multi-scale token aggregation. In Proceedings of the IEEE/CVF Conference on Computer Vision and Pattern Recognition (pp. 10853-10862).

**Questions:**

Please refer weakness

---

> ### Author Response · Authors · 2023-11-13
> **Responses to Reviewer 3 #part.1**
>
> We express our heartfelt gratitude to the reviewers for their excellent evaluation of our paper’s Presentation and for their recognition of the contribution and potential impact of Length-scaled cosine attention. This serves as a significant encouragement for our work that has spanned over a year.
>
> **R1: Novelty and Distinction from Previous Work**
>
> We highly appreciate the insightful comments from the reviewers that the concept of queries attending to both fine-grained and coarse-grained information simultaneously has been extensively studied previously. This indicates that our previous description of our contributions did not adequately express the innovative aspects of our work compared to previous studies. We have revised these descriptions in the new version of our paper.
>
> We believe that the contribution and novelty of our work lie in proposing an attention mechanism that aligns more closely with biological foveal vision than previous work. Its pixel-wise operation can effectively simulate the continuous movement of the eyeball, while each pixel possesses a comprehensive global perception.
>
> In contrast to our approach, Focal Transformer still operates on a window-wise manner. The visual perception of queries located at the window’s edge is unnatural compared to biological foveal vision. They cannot maintain fine-grained attention to all their nearest neighbor tokens, and fine-grained features are not centered on the query at the window’s edge. Imagining the effect of changing the black area of Local Attention in Fig.1 of our paper to a blurred area might be helpful. We believe that this trace of window partitioning could introduce some artifacts into the model’s perception.
>
> Furthermore, the CF-VIR paper recommended by the reviewer is commendable. They further subdivide the image patches of the main part of the picture into fine-grained divisions. Although this combination of coarse-grained and fine-grained does not involve the concept of biological foveal vision, it is still an enlightening method. We would like to express our sincere gratitude to the reviewer for recommending relevant papers.
>
> The motivation for proposing PFA is that previous work and experiments have found that deep networks with residual blocks behave like ensembles of shallower networks. This implies that cross-layer information exchange is not as effective as expected. In convolutional networks, this is manifested as the receptive field of deep networks based on stacked convolutional kernels not being as large as expected (hence the proposal of the super-large convolutional kernel scheme). So, what about ViT models? We found that in various efficient ViT models, the information exchange between cross-window or global and local features formed by stacking blocks is not as sufficient as expected. This can be clearly observed in our Appendix E based on Effective Receptive Field (ERF) visualization. Models based on local windows (Swin, CSWin) present unique blocky patterns in visual perception, and these pattern styles are closely related to their window design methods. This indicates that even after many layers of stacking, the exchange of information across windows is still insufficient, resulting in unnatural information mixing. Therefore, we believe that it is very important to implement a pixel-wise global-local perception mode that is closer to biological vision and can simulate eyeball movement in a single token mixer, so we proposed PFA and subsequent Aggregated Attention. The visualization based on ERF shows that our method forms a very natural and smooth information perception mode.
>
> Regrettably, the official model weights for the Focal-Transformer are no longer accessible, and there is a lack of third-party backups. Consequently, we were unable to include it in the ERF visualization comparison. As a compensation, we have visualized the ERF of CSWin, recommended by the reviewer, in our latest revision. It can be observed that in Stage 3, even after 21 layers of stacking, CSWin still exhibits its unique cross-shaped pattern.
>
> Moreover, in Appendix B.2, we conducted a comparison with the super-large convolutional kernel scheme. It can be seen that our model has more advantages in large-scale image inference than RepLKNet(CVPR 2022) and SLaK(ICLR 2023), while the latter two show some limitations in large-size image inference. We believe that this comparative experiment is enlightening for the subsequent research of the computer vision community in related fields.

---

> ### Author Response · Authors · 2023-11-13
> **Responses to Reviewer 3 #part.2**
>
> **R2 Ablation Experiment of Replacing the Baseline Model**
>
> We fully understand the reviewer’s concern about using PVTv2 as a potentially weak baseline. We used CSwin-T to change the token mixer of stages 1-3 to PFA and changed the token mixer of the fourth stage to MHSA (using the P-P-P-M structure to keep consistent with our TransNeXt’s A-A-A-M). Under the condition that the dimensions, structures, and other aspects of the model remain consistent with CSWin-T, we conducted a direct comparison with CSWin on ImageNet-1K. After the replacement, it can be observed that our PFA method can improve the accuracy by 0.5%.
>
> | Models | Channels|Blocks|Params(M) | FLOPs（G） |Top-1 (%)|
> | --- | --- | --- | --- |--- |--- |
> |CSWin|[64,128.256,512]|[1,2,21,1]|23|4.3|82.8|
> |CSWin-PFA|[64,128.256,512]|[1,2,21,1]|24|4.8|83.3|
>
> We once again express our gratitude to the reviewers for their meticulous review and insightful suggestions regarding our paper.

---

### Official Review · Reviewer_4sZd · 2023-10-31

**Soundness:** 3 good
**Presentation:** 3 good
**Contribution:** 2 fair
**Rating:** 5
**Confidence:** 4

**Summary:**

The author introduces a novel attention mechanism, pixel-focused attention (PFA), inspired by biomimetic design principles. This mechanism effectively captures both fine-grained local and coarse-grained global features, eliminating the need for alternately stacking token mixers or incorporating convolution in attention operations, as commonly done in existing methods. Building upon PFA, the author introduces enhanced modules, such as Conv-GLU, Learnable LKV, QLV, and others, to establish the new ViT backbone, TransNeXt, tailored for visual tasks. The effectiveness of TransNeXt is substantiated through comprehensive experiments.

**Strengths:**

- The paper is technically sound;
- The representation of the paper is good;
- The experiments conducted are comprehensive and fully validate the effectiveness of the proposed method.

**Weaknesses:**

My primary reservation regarding the acceptance of this paper pertains to its limited novelty. As depicted in Table 11, many of the modules or concepts presented in the paper have previously been explored in existing research. For instance, the approach to fine-grained local features and coarse-grained global features rooted in biological visual design has been introduced by Focal-Transformer. The non-QKV strategy has already been employed in works like Involution and VOLO. Additionally, the ConvGLU module seems to be a marginal enhancement to the existing blocks illustrated in Figure 3. While I recognize that there might be subtle, optimized implementation details unique to this paper, the overarching narrative gives the impression that TransNeXt is primarily an assembly of modules sourced from other research papers.

**Questions:**

See weakness part.

**Details Of Ethics Concerns:**

N.A.

---

> ### Author Response · Authors · 2023-11-13
> **Responses to Reviewer 2 #part.1**
>
> We are deeply appreciative of the reviewers for their meticulous reading of our paper. We are gratified that they acknowledged the representation of our paper and deemed our experiments comprehensive, fully validating the effectiveness of our proposed method. We understand the reviewers’ concerns about the novelty of our method, which is a valuable insight. This indicates that our writing did not effectively reflect the novelty of our method compared to previous work, causing confusion for the reviewers. We have revised these descriptions in the updated version of our paper.
>
> Our manuscript and model encapsulate a series of innovative points accomplished over the span of more than a year. In essence, our paper is a comprehensive redesign of the original baseline, integrating these innovative points. In the review process, we found that each reviewer acknowledged the novelty and contribution of different components. Reviewer 1 recognized the novelty of the PFA module, Reviewer 3 acknowledged the contribution of Length-scaled cosine attention in multi-scale inference, and Reviewer 4 recognized the PFA module as "a notable aspect" and Convolutional GLU as "a valuable element". We express our sincere gratitude to the reviewers for their recognition of these contributions.
>
> Therefore, we kindly request the opportunity to elaborate on the novelty of our work and its distinction from previous works in the following sections.
>
> **R1: Novelty and Motivation of FPA and its Distinction from Previous Work**
>
> We believe that the contribution and novelty of our work lie in proposing an attention mechanism that aligns more closely with biological foveal vision than previous work. Its pixel-wise operation can effectively simulate the continuous movement of the eyeball, while each pixel possesses a comprehensive global perception.
>
> In contrast to our approach, Focal Transformer still operates on a window-wise manner. The visual perception of queries located at the window’s edge is unnatural compared to biological foveal vision. They cannot maintain fine-grained attention to all their nearest neighbor tokens, and fine-grained features are not centered on the query at the window’s edge. Imagining the effect of changing the black area of Local Attention in Fig.1 of our paper to a blurred area might be helpful. We believe that this trace of window partitioning could introduce some artifacts into the model’s perception.
>
> The motivation for proposing PFA is that previous work and experiments have found that deep networks with residual blocks behave like ensembles of shallower networks. This implies that cross-layer information exchange is not as effective as expected. In convolutional networks, this is manifested as the receptive field of deep networks based on stacked convolutional kernels not being as large as expected (hence the proposal of the super-large convolutional kernel scheme). So, what about ViT models? We found that in various efficient ViT models, the information exchange between cross-window or global and local features formed by stacking blocks is not as sufficient as expected. This can be clearly observed in our Appendix E based on Effective Receptive Field (ERF) visualization. Models based on local windows (Swin, CSWin) present unique blocky patterns in visual perception, and these pattern styles are closely related to their window design methods. This indicates that even after many layers of stacking, the exchange of information across windows is still insufficient, resulting in unnatural information mixing. Therefore, we believe that it is very important to implement a pixel-wise global-local perception mode that is closer to biological vision and can simulate eyeball movement in a single token mixer, so we proposed PFA and subsequent Aggregated Attention. The visualization based on ERF shows that our method forms a very natural and smooth information perception mode.
>
> Regrettably, the official model weights for the Focal-Transformer are no longer accessible, and there is a lack of third-party backups. Consequently, we were unable to include it in the ERF visualization comparison. As a compensation, we have visualized the ERF of CSWin in our latest revision. It can be observed that in Stage 3, even after 21 layers of stacking, CSWin still exhibits its unique cross-shaped pattern.
>
> Moreover, in Appendix B.2, we conducted a comparison with the super-large convolutional kernel scheme. It can be seen that our model has more advantages in large-scale image inference than RepLKNet(CVPR 2022) and SLaK(ICLR 2023), while the latter two show some limitations in large-size image inference. We believe that this comparative experiment is enlightening for the subsequent research of the computer vision community in related fields.

---

> > ### Author Response · Authors · 2023-11-13
> > **Responses to Reviewer 2 #part.2**
> >
> > **R2: The novelty and efficiency of our method to unify QKV, LKV, and QLV mechanisms**
> >
> > The main concern raised in the review comments was that the introduction of QLV and LKV attention operations in PFA made our method appear to be a combination of existing technologies. We believe that such concerns are reasonable and insightful. This indicates that our paper did not effectively explain the advantages and competitiveness of our method of integrating these three mechanisms. We have revised the relevant statements in the latest revision. Our integration method requires very little additional overhead, is highly efficient, and has novelty in the way it is integrated.
> >
> > Since the introduction of the Non-QKV mechanism in the Synthesizer(ICML 2021) paper, works such as QnA (CVPR 2022) and VOLO(TPAMI 2023)/Involution(CVPR 2021) have validated the feasibility of LKV and QLV mechanisms in visual models. We note that no previous work has attempted to unify QKV, LKV, and QLV, these three attention mechanisms, in a single attention layer (we believe that merely implementing through block stacking or ensemble methods would be trivial and lack novelty, as these methods would also be affected by the potential depth degradation of residual connections). We consider this work to be the first attempt to unify these three attention mechanisms in a single attention layer.
> >
> > Our design of PFA serves as a promising foundation for unifying these three attention mechanisms. It inherently has a sliding window attention branch, and reusing this branch reduces the overhead required by the QLV mechanism to $HWk^2C$. Our method of introducing the LKV mechanism is more efficient, requiring only the addition of a query embedding to all queries to achieve the sum of the LK affinity matrix and the QK affinity matrix, with its additional overhead being negligible. In terms of experimental results, enhancing PFA to Aggregated Attention requires only about **0.2%(of base model) to 0.3%(of  micro model)** of the additional computational consumption in the entire model, but the improvement is significant, achieving a very cost-effective trade-off. From this perspective, our method of unifying QKV, QLV, and LKV attentions does not employ a trivial approach, and is both efficient and has its innovative aspects. We believe this is a successful attempt.
> >
> > | Models | Params | FLOPs（G） | IN-1K | IN-A |
> > | --- | --- | --- | --- | --- |
> > |TransNeXt-Micro(FPA)|12.78|2.65|81.8|26.9|
> > |TransNeXt-Micro(AA)|12.81 (+0.2%)|2.66(+0.3%)|82.5(+0.7%)|29.9(+3.0%)|
> >
> >
> > We once again express our gratitude to the reviewers for their meticulous review and insightful suggestions regarding our paper.

---

### Official Review · Reviewer_hJkq · 2023-11-01

**Soundness:** 3 good
**Presentation:** 3 good
**Contribution:** 1 poor
**Rating:** 3
**Confidence:** 4

**Summary:**

This paper proposes a novel attention mechanism, several new components, and a transformer architecture. The motivation for the attention mechanism is combining global perception with local recognition. Reasonable results are reported.

**Strengths:**

1. Some of the proposed components are novel, including PFA and Activate and Pool.

2. Though some other proposed structures are nothing interesting, they work well (e.g., ConvGLU).

**Weaknesses:**

1. Some claims are inappropriate or wrong

1.1 [it attains a box mAP of 55.1 using the DINO detection head, outperforming ConvNeXt-L ...] They are not comparable. ConvNeXt used UPerNet.

1.2 [This is the first token mixer that simultaneously satisfies fine-grained perception near the focus, coarse-grained global perception at a distance, and pixel-wise translational equivariance] Very large (larger than 51x51) and sparse convolution is exactly a token mixer with such properties so it is inappropriate to claim the proposed token mixer as the first. The magnitudes of outer parameters of a very large convolution kernel are small and sparse while the central parameters are dense. Please refer to the paper of [SLaK].

1.3 [More elegant design] Compared to what? Is adding a depthwise 3x3 elegant?

2. I seriously doubt the efficiency of the proposed structure. It is too complicated and the implementation of PFA may require naive indexing operations, which are extremely inefficient. The actual throughput and latency test results and the comparisons with other competitors are missing, which is unacceptable.

3. I admit PFA is novel but do not take Aggregated Attention as a significant contribution since Aggregated Attention = PFA + query embedding + positional attention, and the latter two are common practices.

In summary, I recommend rejecting this paper because it reads like yet another customized attention (which is neither simple nor efficient) plus some common practices borrowed from other works. And though the results on ImageNet-1K look promising, no results with larger models nor bigger data are reported.

**Questions:**

It is claimed that the proposed pixel-focused attention "possesses visual priors comparable to convolution." So why not just use convolution? Discussions and comparisons are missing.

---

> ### Author Response · Authors · 2023-11-13
> **Responses to Reviewer 1**
>
> We are deeply appreciative of the reviewers’ meticulous reading of our paper, and we find many of their suggestions to be insightful and helpful. We are grateful for the reviewers’ recognition of the novelty of our proposed Pixel-Focused Attention (PFA) and Activate and Pool modules.
>
> **Q1:** Why not use convolution? Why propose pixel-focused attention “possesses visual priors comparable to convolution”?
>
> We appreciate the reviewer's insightful question, which touches upon the underlying motivation for our Pixel-Focused Attention (PFA) design.
>
> The primary motivations for proposing pixel-focused attention are twofold:
> 1. To circumvent the depth degradation resulting from the stacking of residual modules, which hampers cross-layer information exchange to a degree less effective than anticipated.
> 2. On the basis of satisfying (1), to propose an token mixer that aligns more closely with the biological vision system, capable of simulating the continuous movement of the eyeball, ensuring that the visual perception of the query at each position on the feature map remains highly consistent with biological foveal vision.
>
> From this perspective, convolution is closer to the working mode of the eyeball than various local window-based attentions (Swin, Focal Transformer), which we believe is the advantage of convolution in visual priors. However, the depth degradation of residual connections affects convolution by making its effective receptive field smaller than expected, hence various large convolution kernel models (RepLKNet(CVPR 2022), SLaK(ICLR 2023)) have been proposed to address this issue. The results under ERF visualization can prove the effectiveness of the large convolution kernel strategy.
>
> However, we observed a significant performance decay in large-scale image inference for these models based on the large convolution kernel strategy pre-trained at a resolution of $224^2$ (in Appendix B.2). The top-1 accuracy of RepLKNet-31B at a resolution of $640^2$ is only 0.9%, which to some extent shows the limitations of the large convolution kernel strategy in broader application scenarios. Therefore, we proposed pixel-focused attention, attempting to approximate the priors of convolution in biomimetic vision from the perspective of the attention mechanism. We proposed length-scaled cosine attention and used a position encoding scheme with extrapolation capability, achieving better large-scale image extrapolation performance than pure convolution networks. PFA and AA also have a linear complexity inference mode comparable to convolution. We compared TransNeXt-T with ConvNeXt-B and RepLKNet-B in Appendix B.2 and discussed this point. In the latest revision, we added SLaK-S as a comparison (thanks again to the reviewer for recommending reading the SLaK paper) Our observations indicate that TransNeXt-T significantly outperforms pure convolution models in large-scale image inference, demonstrating that this is another effective approach to addressing the two motivations mentioned above. Furthermore, the performance degradation of large convolution kernel models such as RepLKNet and SLaK, as reported in our experiments during large-scale image inference, warrants attention from the research community. These findings may contribute to future research in this field.
>
> **R1.2:** The statement “This is the first token mixer that…” is inappropriate.
>
> We appreciate the reviewer’s careful reading. Indeed, we overlooked the blurring effect caused by the sparsity of the large convolution kernel periphery during our writing. Therefore, such a statement is not rigorous within the scope of the token mixer. We have used a more precise description in the new revision.
>
> **R1.1:** "It attains a box mAP of 55.1 using the DINO detection head, outperforming ConvNeXt-L …" They are not comparable. ConvNeXt used UPerNet.
>
> We value the reviewer’s attention to data details, which is commendable. Indeed, the ConvNeXt paper did not provide results for the DINO detection head. Our data references the results of the open-source weight report from the detrex project: https://github.com/IDEA-Research/detrex/tree/main/projects/dino#pretrained-dino-with-convnext-backbone . The box map of DINO-ConvNeXt-Large-384-4scale using IN-1K is 53.4, which is correct. For Swin using the DINO detection head, we referenced the open-source weight results from mmdet as much as possible, as we also used mmdet for training, to maintain consistency.

---

> ### Author Response · Authors · 2023-11-13
> **Responses to Reviewer 1 #part.2**
>
> **R1.3:** [More elegant design] Compared to what? Is adding a depthwise 3x3 elegant?
>
> We appreciate the reviewer’s inquiry into the details of our paper. The term “more elegant design” refers to our ConvGLU in comparison to the Feedforward using the SE mechanism. We have noticed that some studies, such as FAN-ViT-SE(ICML 2022) and MaxViT(ECCV 2022) (used in their MBConv block), still employ the SE mechanism to enhance channel attention in models, thereby improving robustness. The SE mechanism uses Global Average Pooling for downsampling, an extremely coarse-grained solution that subjects all pixel tokens on the feature map to the same gating signal, regardless of their individual information. In contrast, we incorporate a 3x3 convolution into GLU without downsampling, allowing each token on the feature map to receive a distinct gating signal. Moreover, these gating signals are generated based on the nearest neighbor features around the token. Compared to the self-gating mode of GLU itself, the gating signals of ConvGLU have a larger perception, hence more effective decision-making can be expected. Additionally, it has fewer FLOPs than ConvFFN, which incorporates convolution into FFN, and it exhibits stronger performance and robustness. It also possesses the ability to serve as position encoding. In Appendix B.4, we explore three alternative methods of integrating 3x3 convolution in GLU, underscoring the significance of the placement of 3x3 convolution.  It can be observed that our ConvGLU design is currently optimal.
>
> In summary, we believe this is a design that can effectively meet the diverse requirements of contemporary ViT and is sufficiently simple.
>
> **R2:** Throughput and Latency Test
>
> Currently, we have only implemented native CUDA code without extensive optimization. Consequently, it is not as efficient as highly optimized dense GPU operators. We tested the model’s throughput on a V100 16G with FP32 precision at a batch size of 64, and the model’s latency at a batch size of 10. Our biomimetic vision implementation based on the sliding window is more efficient than the Focal Transformer method. Moreover, compared to previous models such as MaxViT (ECCV 2022), QuadTree (ICLR 2022), and BiFormer (CVPR 2023), our model achieved competitive TOP-1 accuracy results under similar throughput conditions.
>
> | Models | Params | FLOPs（G） | TOP-1 | throughputs(img/s) | latency(ms) |
> | --- | --- | --- | --- | --- | --- |
> | BiFormer-T | 13.1 | 2.2 | 81.4 | 828 | 1.73 |
> | Swin-T | 28.3 | 4.5 | 81.2 | 790 | 1.44 |
> | ConvNeXt-T | 28.6 | 4.5 | 82.3 | 779 | 1.42 |
> | QuadTree-B-b1 | 13.6 | 2.3 | 80.0 | 663 | 2.85 |
> | **TransNeXt-Micro** | 12.8 | 2.7 | **82.5** | 641 | 3.95 |
> | Swin-S | 49.6 | 8.7 | 83.1 | 460 | 2.51 |
> | MaxViT-Tiny | 30.9 | 5.6 | 83.4 | 459 | 2.77 |
> | ConvNeXt-S | 50.2 | 8.7 | 83.1 | 441 | 2.54 |
> | **TransNeXt-Tiny** | 28.2 | 5.7 | **84.0** | 413 | 3.98 |
> | BiFormer-S | 25.5 | 4.5 | 83.8 | 396 | 3.62 |
> | QuadTree-B-b2 | 24.2 | 4.5 | 82.7 | 361 | 5.51 |
> | Focal-Transformer-T | 29.1 | 4.9 | 82.2 | 337 | 3.3 |
> | Swin-B | 87.8 | 15.4 | 83.5 | 292 | 3.73 |
> | ConvNeXt-B | 88.6 | 15.4 | 83.8 | 290 | 3.8 |
> | MaxViT-Small | 68.9 | 11.7 | 84.4 | 273 | 4.2 |
> | BiFormer-B | 56.8 | 9.8 | 84.3 | 241 | 4.7 |
> | QuadTree-B-b3 | 46.3 | 7.8 | 83.7 | 238 | 8.9 |
> | **TransNeXt-Small** | 49.7 | 10.3 | **84.7** | 214 | 6.85 |
> | Focal-Transformer-S | 51.1 | 9.1 | 83.5 | 203 | 6.46 |
> | QuadTree-B-b4 | 64.2 | 11.5 | 84.0 | 166 | 13.17 |
> | **TransNeXt-Base** | 89.7 | 18.4 | **84.8** | 151 | 7.38 |
> | Focal-Transformer-B | 89.8 | 16 | 83.8 | 145 | 7.63 |
> | MaxViT-Base | 119.5 | 24 | 84.9 | 144 | 8.13 |
>
> Furthermore, the speed of TransNeXt is expected to improve with more engineering efforts. We will continue to provide more efficient operator optimizations in the future to enhance the competitiveness of TransNeXt.

---

> ### Author Response · Authors · 2023-11-13
> **Responses to Reviewer 1 #part.3**
>
> **R3:** PFA is novel, but Aggregated Attention = PFA + query embedding + positional attention.
>
> We would like to express our gratitude once again to the reviewers for recognizing the novelty of our Pixel-Focused Attention (PFA) module. We believe that such concerns are reasonable and insightful. This indicates that our paper did not effectively explain the advantages and competitiveness of our method of integrating these three mechanisms. We have revised the relevant statements in the latest revision. Our integration method requires very little additional overhead, is highly efficient, and has novelty in the way it is integrated.
>
> Since the introduction of the Non-QKV mechanism in the Synthesizer(ICML 2021) paper, works such as QnA (CVPR 2022) and VOLO(TPAMI 2023)/Involution(CVPR 2021) have validated the feasibility of LKV and QLV mechanisms in visual models. We note that no previous work has attempted to unify QKV, LKV, and QLV, these three attention mechanisms, in a single attention layer (we believe that merely implementing through block stacking or ensemble methods would be trivial and lack novelty, as these methods would also be affected by the potential depth degradation of residual connections). We consider this work to be the first attempt to unify these three attention mechanisms in a single attention layer.
>
> Our design of PFA serves as a promising foundation for unifying these three attention mechanisms. It inherently has a sliding window attention branch, and reusing this branch reduces the overhead required by the QLV mechanism to $HWk^2C$. Our method of introducing the LKV mechanism is more efficient, requiring only the addition of a query embedding to all queries to achieve the sum of the LK affinity matrix and the QK affinity matrix, with its additional overhead being negligible. In terms of experimental results, enhancing PFA to Aggregated Attention requires only about **0.2%(of base model) to 0.3%(of  micro model)** of the additional computational consumption in the entire model, but the improvement is significant, achieving a very cost-effective trade-off. From this perspective, our method of unifying QKV, QLV, and LKV attentions does not employ a trivial approach, and is both efficient and has its innovative aspects. We believe this is a successful attempt.
>
> | Models | Params | FLOPs（G） | IN-1K | IN-A |
> | --- | --- | --- | --- | --- |
> |TransNeXt-Micro(FPA)|12.78|2.65|81.8|26.9|
> |TransNeXt-Micro(AA)|12.81 (+0.2%)|2.66(+0.3%)|82.5(+0.7%)|29.9(+3.0%)|
>
> We once again express our gratitude to the reviewers for their meticulous review and insightful suggestions regarding our paper.

---

### Author Response · Authors · 2023-11-13
**Paper Update**

We extend our gratitude to all reviewers for their meticulous review of our paper and for providing insightful suggestions and comments. Herein, we report the revisions made to our paper:

1. We have revised the introduction to strengthen the motivation for the Pixel-focused Attention (PFA) module and have articulated our contributions more accurately.
2. We have included the multi-scale evaluation results of UperNet under the extrapolation strategy in Table 14 and corrected the data entry error for TransNeXt-Base (the correct value is slightly better).
3. We have incorporated SLaK into the multi-scale image inference comparison in Appendix B.2 (We appreciate Reviewer 1 for recommending this work), and emphasized the advantages of our approach over large convolution kernel schemes in addressing depth degradation in the main text.
4. We have added the visualization results of CSWin in the comparison based on Effective Receptive Field in Appendix E.